# Self-Bootstrapping for Versatile Test-Time Adaptation

Shuaicheng Niu [* 1 2]   Guohao Chen [* 1]   Peilin Zhao [3]   Tianyi Wang [1]   Pengcheng Wu [1]   Zhiqi Shen [† 1]

## Abstract

In this paper, we seek to develop a versatile test-time adaptation (TTA) objective for a variety of tasks — classification and regression across image-, object-, and pixel-level predictions. We achieve this through a self-bootstrapping scheme that optimizes prediction consistency between the test image (as target) and its deteriorated view. The key challenge lies in devising effective augmentations/deteriorations that: i) preserve the image's geometric information, *e.g.*, object sizes and locations, which is crucial for TTA on object/pixel-level tasks, and ii) provide sufficient learning signals for TTA. To this end, we analyze how common distribution shifts affect the image's information power across spatial frequencies in the Fourier domain, and reveal that low-frequency components carry high power and masking these components supplies more learning signals, while masking high-frequency components can not. In light of this, we randomly mask the low-frequency amplitude of an image in its Fourier domain for augmentation. Meanwhile, we also augment the image with noise injection to compensate for missing learning signals at high frequencies, by enhancing the information power there. Experiments show that, either independently or as a plug-and-play module, our method achieves superior results across classification, segmentation, and 3D monocular detection tasks with both transformer and CNN models.

## 1. Introduction

Despite deep learning evolving at an incredible speed, its ability to generalize to out-of-distribution (OOD) domains remains a long-existed challenge (Hendrycks & Dietterich, 2019; Koh et al., 2021), drawing substantial interest from both research and industry. To address this, numerous methods have been explored, including *training-time* approaches like domain generalization (Shankar et al., 2018; Dou et al., 2019) and data augmentation (Hendrycks et al., 2020; Yao et al., 2022), as well as *test-time* techniques such as source-free domain adaptation (Qiu et al., 2021) and test-time adaptation (TTA) (Liang et al., 2023), to name just a new.

Among these approaches, TTA (Sun et al., 2020; Niu et al., 2023; Iwasawa & Matsuo, 2021; Bartler et al., 2022; Liang et al., 2023; Wang et al., 2024) has emerged as a rapidly advancing research area. By adapting to each data point immediately after inference, TTA achieves minimal overhead for model updates, making it highly appealing to a broad audience in real-world applications. However, current TTA solutions remain limited in scope, supporting only a narrow range of tasks due to constraints in their underlying self-supervised learning objectives, as depicted below.

Self-learning/entropy-based TTA methods (Wang et al., 2021; Niu et al., 2022a; 2023) and prototype-based methods (Iwasawa & Matsuo, 2021) focus on minimizing the entropy of predicted logits or maintaining class-wise prototypes for discriminative models, *e.g.*, image classification, making them unsuitable for regression tasks. Alignment-based approaches (Mirza et al., 2023; Lin et al., 2023) conduct adaptation by aligning feature statistics, *e.g.*, mean and variance, between the target and source data. While this approach is more general and applicable to a broader of tasks, it requires pre-calculating source statistics with access to source data, which raises data privacy concerns (Liang et al., 2020; Wang et al., 2021). Moreover, this method regularizes statistics to align with the source but overlooks direct learning from test data, leading to limited performance in more complex scenarios, *see* Tables 2 and 3.

Consistency-based methods (Zhang et al., 2022; Shu et al., 2022) are another major category of TTA, such as MEMO (Zhang et al., 2022), which optimizes prediction consistency across different augmented views of the input image. Their augmentation strategy typically follows the well-established practice in contrastive learning methods like MoCo (He et al., 2020) and SimCLR (Chen et al., 2020a), relying on techniques such as random cropping

[*]Equal contribution [†]Corresponding author [1]College of Computing and Data Science, Nanyang Technological University, Singapore [2]Joint WeBank-NTU Research Institute on Fintech, Singapore [3]School of Artificial Intelligence, Shanghai Jiao Tong University, China. Correspondence to: <shuaicheng.niu@ntu.edu.sg>.

*Proceedings of the $42^{nd}$ International Conference on Machine Learning*, Vancouver, Canada. PMLR 267, 2025. Copyright 2025 by the author(s).

and resizing. However, these augmentations can disrupt the overall content of the image, *i.e.*, the image's geometric structure—objects sizes, locations, relative layouts, and *etc*. While this may not impact image-level predictions like classification, it shall fail on more fine-grained tasks, such as object detection, which require the precise predictions of each object's coordinates and sizes.

In this paper, we aim to develop a new fully TTA method to adapt an arbitrary trained model, while requiring no access to source data or altering the original training process. We build on the general idea of consistency-based learning, but extend it to a more unified, architecture-agnostic, and task-agnostic self-bootstrapping TTA approach, so that it is applicable to classification and regression tasks across image-, object-, and pixel-level prediction models.

To be specific, in our **s**elf-bootstra**p**ping TT**A** (SPA) framework, we use the predictions of the original image (strong view) as the target, which provide supervision to guide the model learning in making consistent predictions on a deteriorated view (weak) of the same image. This process enhances the predictions of the weak image view, which, in turn, feeds back and improves the original predictions through shared model parameters. Here, the key challenge of making this self-bootstrapping learning scheme applicable to fine-grained object- or pixel-level tasks, is designing effective augmentations—which need to preserve the main image's geometric structure while introducing sufficient differences (learning signals)—to deteriorate a given image.

To the above end, we propose randomly masking amplitudes in the image's Fourier frequency domain for augmentation. Specifically, we analyze how common distribution shifts manifest in the frequency domain by comparing the radially averaged power spectral density (RAPSD) (Van der Schaaf & Van Hateren, 1996) of shifted and source domains in Figure 2, where RAPSD reflects image information power across spatial frequencies. We observe that original images typically exhibit low RAPSD (*i.e.*, low information power) at high frequencies, with several domain shifts further reducing it. Thus, masking high-frequency amplitudes tends to provide limited learning signals (*see* Table 6). In contrast, images show high RAPSD at low frequencies, indicating that there will be a larger RAPSD difference before and after masking there, and thus can provide richer learning signals. Therefore, we only mask the low-frequency component of the amplitude. Moreover, to compensate for the lack of learning signals at high frequencies, we augment the image by injecting random Gaussian noise into it, which enhances RAPSD in the high-frequency range, thereby supplying learning signals across all frequencies. Lastly, to make SPA more stable and reliable, we introduce an active self-bootstrapping learning scheme—for tasks including classification heads, we perform adaptation only when the

model has higher prediction confidence on the target strong image view than on the deteriorated weak image view.

**Main Novelty and Contributions**  **1)** We propose a simple yet effective active self-bootstrapping learning framework for TTA, which is general to be used for classification and regression across image/object/pixel-level tasks, showing broad applicability. **2)** We analyze how common domain shifts manifest in the Fourier frequency domain and, based on this analysis, propose geometry-preserving augmentations—low-frequency amplitude masking and high-frequency noise injection. These augmentations supply learning signals for our self-bootstrapping adaptation across all spatial frequencies, significantly enhancing adaptation performance. **3)** Extensive experiments across classification, segmentation, and 3D monocular detection with both transformer and CNN models demonstrate our superiority.

## 2. Preliminary and Problem Statement

We briefly revisit fully TTA in this section for the convenience of our method presentation and put **detailed related work discussions into Appendix** A due to page limits.

**Fully Test-Time Adaptation (TTA)**  In this paper, we focus on the problem of fully TTA. Formally, given any model $f(\cdot; \theta)$ trained on source data $\mathcal{D}_{train} = \{(\mathbf{x}_i, y_i)\}_{i=1}^N$, fully TTA adapts $f(\cdot; \theta)$ to testing data $\mathcal{D}_{test} = \{\mathbf{x}_j\}_{j=1}^M$ with potential distributions shifts from $\mathcal{D}_{train}$ on the fly using some unsupervised learning objectives (Wang et al., 2021) $\mathcal{L}$, *i.e.*, $\min_{\tilde{\theta}} \mathcal{L}(f(\mathbf{x}; \theta))$, where $\tilde{\theta} \subset \theta$ are learnable parameters during TTA. Here, test samples arrive in an online data stream. The fully TTA process does not alter the original model training process or require access to source data, making it practical and easy to implement in real-world applications.

**Problem Statement and Motivation**  Existing fully TTA methods often suffer a limited application scope and are not general enough. For instance, entropy-based methods (Wang et al., 2021; Niu et al., 2022a) are restricted to classification and not compatible with regression tasks. Augmentation consistency-based methods (Zhang et al., 2022) work well for image-level recognition but may tend to fail in object- and pixel-level prediction tasks, *e.g.*, object detection, where precise coordinates and dimensions of objects shall be destroyed by their commonly used augmentations like random cropping and resizing. In real-world applications, tasks are typically various, and selecting or redesigning working TTA solutions for different tasks or models can be inconvenient or impractical. Therefore, in this paper, we aim to develop a more versatile fully TTA method that is task-agnostic, supporting both classification and regression across image-, object- and pixel-level prediction tasks.

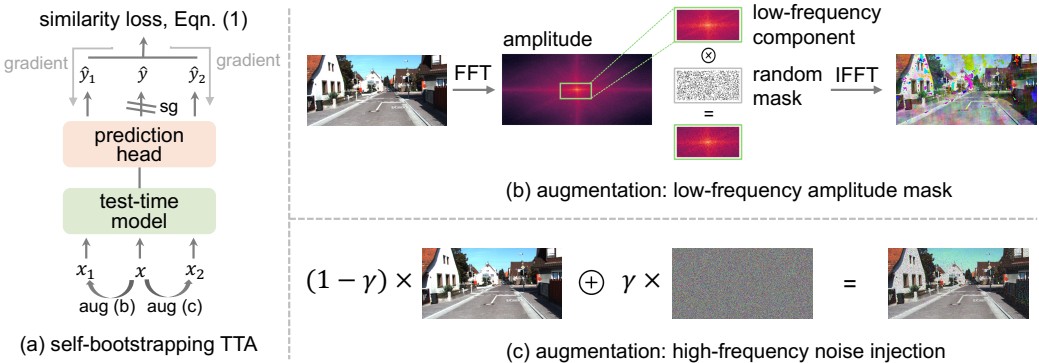

*Figure 1.* Illustration of SPA method. **(a)** We conduct self-bootstrapping learning for TTA by maximizing prediction consistency from the weak augmented/deteriorated views to the strong original image view. The augmentations are designed to preserve geometric structure by **(b)** randomly masking low-frequency components of the image's amplitude in the Fourier domain, and **(c)** injecting Gaussian noise into the original image to enhance the information intensity on high frequency. 'sg': stop gradient. (I)FFT: (Inverse) Fast Fourier Transform.

## 3. Approach

We achieve the goal of task-agnostic and architecture-agnostic versatile TTA by establishing a framework of **s**elf-bootstra**p**ping **a**daptation with geometric structure-preserving weak-to-strong learning, *namely* SPA. Here, we begin with self-bootstrapping, which introduces a general idea of refining the model predictions, using its own outputs as training targets. To broaden its compatibility with object-/pixel-level downstream tasks requiring dense prediction, we propose to optimize the prediction consistency between the original image (strong), as targets, and a corresponding geometric structure-preserving deteriorated image (weak). In SPA, we aim to create a weak image view that loses as much information as possible—providing sufficient signals for self-bootstrapping learning, while preserving the overall structural content of the image, such as size and relative location—making it suitable for finer object/pixel-level tasks. We depict the self-bootstrapping scheme in Sect. 3.1 and geometry-preserving augmentations in Sect. 3.2. The overall details of SPA are illustrated in Figure 1 and Algorithm 1.

### 3.1. Active Self-Bootstrapping Learning

We design SPA with inspiration from prior self-supervised contrastive methods like BYOL (Grill et al., 2020) and DINO (Caron et al., 2021), as their core learning scheme, *bootstrapping*, is not limited to classification or regression. Unlike BYOL or DINO which maximize consistency between two randomly augmented views for representation learning, we extend this framework for TTA by introducing weak-to-strong learning, *i.e.*, from a randomly deteriorated view (weak) to the original image (strong). This adjustment is necessary because, in TTA, we are adapting a well-trained model to new, out-of-distribution domains where reliable

signals are essential for guiding the prediction adaptation; otherwise, it risks degrading the model's performance.

To make this self-bootstrapping TTA applicable to object-/pixel-level tasks, we define geometric structure-preserving random augmentations $\mathcal{T}$. This differs from heavy geometric augmentations required by prior self-supervised methods, such as random cropping and resizing, which significantly alter pixel-level content (*e.g.*, object locations and sizes) from the original image, making them unsuitable in our weak-to-strong self-bootstrapping learning. For presentation coherence, we leave the augmentation details in Sect. 3.2. Given test image $\mathbf{x}$, SPA creates an augmented view $\mathbf{v} = t(\mathbf{x})$ with $t \sim \mathcal{T}$. We take $\mathbf{v}$ as the weak view and the original $\mathbf{x}$ as the strong view. We then minimize a similarity loss $\mathcal{L}_s$ between the predictions of $\mathbf{x}$ and $\mathbf{v}$ using a confidence-aware selection function $S(\mathbf{x}, \mathbf{v})$ (for tasks including classification heads), which actively determines whether to perform optimization w.r.t. a given sample or pixel to mitigate the influence of unreliable supervisions. Formally, our self-bootstrapping TTA formula is given by:

$$\min_{\tilde{\theta}} S(\mathbf{v}, \mathbf{x})\mathcal{L}_s\big(f(\mathbf{v};\theta),\ f(\mathbf{x};\theta)\big), \mathbf{v} = t(\mathbf{x}),\ \ t \sim \mathcal{T}. \quad (1)$$

Here, $\tilde{\theta} \subset \theta$ denotes the learnable model parameters. For the similarity loss $\mathcal{L}_s(\cdot, \cdot)$, we adopt KL divergence for classification heads and L1 loss for regression heads. We also insert a new learnable projector before the final prediction head, following BYOL (Grill et al., 2020), to prevent the model from converging to trivial solutions in the image-level classification task. This projector is initialized as identity mappings to enable a warm start for the fully TTA process.

The confidence-aware selection $S(\mathbf{v}, \mathbf{x})$ in Eqn. (1) is designed for when the given task comprises classification heads. For image classification tasks, let $\hat{f}(\mathbf{x}, \theta) =$

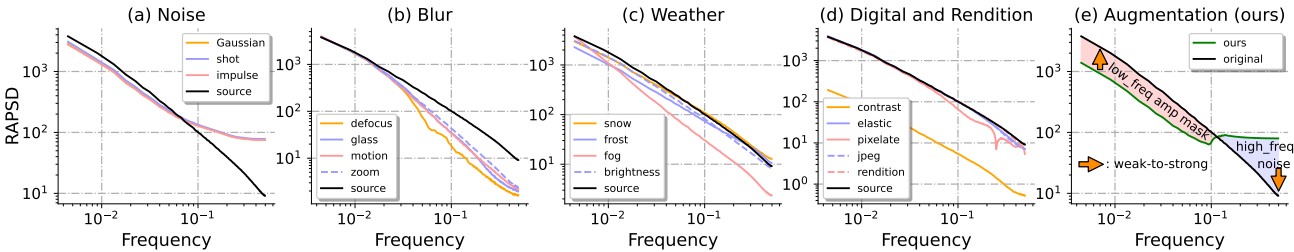

*Figure 2.* **(a-d)** Changes of radially averaged power spectral density (RAPSD) (Van der Schaaf & Van Hateren, 1996) under domain shifts. **(e)** SPA 's geometry-preserving augmentations reduce the RAPSD at low frequencies and enhance it at high frequencies to create deteriorated images for our self-bootstrapping learning. We separately select 512 images from the Source, ImageNet-R, ImageNet-C (15 corruptions), to perform FFT, and visualize their mean RAPSD based on the spectrum amplitude.

$\max(f(\mathbf{x}, \theta))$ and $\mathbb{I}_{\{\cdot\}}(\cdot)$ be an indicator function, $S(\mathbf{v}, \mathbf{x})$ aims to select samples whose prediction confidence of the strong view is higher than the weak view for optimization:

$$S(\mathbf{v}, \mathbf{x}) = \mathbb{I}_{\{\hat{f}(\mathbf{x};\theta) > \hat{f}(\mathbf{v};\theta)\}}(\mathbf{v}, \mathbf{x}), \qquad (2)$$

For segmentation and object detection tasks, the predictions are often made on the pixel-level $\hat{f}(\mathbf{x}/\mathbf{v}; \theta)_{i,j}$. In this case, we conduct selection at the pixel level, and calculate overall similarity loss $\mathcal{L}_s(\cdot, \cdot)$ by averaging it on all selected pixels.

### 3.2. Geometry-Preserving Augmentation

The core idea in SPA is to devise image geometric structure-preserving augmentations to enable self-bootstrapping learning on finer object-/pixel-level tasks. To this end, one can directly exploit conventional image structure-preserving augmentations, *e.g.*, contrast adjustment, brightness changes, and grayscale conversion. However, as in Table 6, these augmentations, either individually or in combination, failed to provide sufficient learning signals, were sensitive to the type of domain shifts, and even led to model collapse.

To ensure the augmentations are geometric-preserving while still providing as much learning guidance as possible for TTA, we propose augmenting the image by randomly masking its amplitude in the Fourier frequency domain. Let $F(\cdot)$ and $F^{-1}(\cdot)$ be the Fast Fourier Transform (FFT) and inverse Fast Fourier Transform (IFFT) operations, respectively. We denote the amplitude and phase component after FFT by $F^A(\cdot)$ and $F^P(\cdot)$. Then, using a random amplitude mask to augment an image $\mathbf{x}$ is defined as

$$\mathbf{v} = F^{-1}([M(m) \circ F^A(\mathbf{x}), \ F^P(\mathbf{x})]). \qquad (3)$$

Here, $\circ$ is element-wise multiplication, $M(m)$ produces a random 0-1 matrix with the same size as $\mathbf{x}$ for masking. The proportion of 0 in $M(m)$ is defined as a mask ratio $m$.

Applying Eqn. (3) to generate a deteriorated view to perform weak-to-strong self-bootstrapping learning in Eqn. (1) already yields much better performance compared to conventional geometry-preserving augmentations that fail to

supply sufficient learning signals, as in Table 6. However, this is still not optimal. To further improve, we first analyze how amplitude changes when domain shift occurs below.

**How Common Domain Shifts Manifest in Frequency Domain?** We compare the radially averaged power spectral density (RAPSD) (Van der Schaaf & Van Hateren, 1996) differences between the out-of-distribution test samples and the in-distribution source samples. Here, RAPSD calculates the average spectral amplitude of an image across a concentric rectangle (within $F^A(\mathbf{x})$ in Eqn. 3) at each frequency, reflecting the distribution of image information power across all frequencies, *i.e.*, the higher RAPSD, the higher information power. From Figure 2 (a-d), common domain shifts typically exhibit the following changes of RAPSD: high-frequency increase (*e.g.*, noise corruptions) or degradation (*e.g.*, blur corruptions), low-frequency degradation (*e.g.*, contrast), or minimal change across all frequencies (*e.g.*, rendition). In the following, we seek to devise general augmentation strategies for various types of domain shifts and based on the observations from Figure 2 (a-d), we draw the following designing motivations.

*1) Low-frequency masking contributes more for* **SPA.** Images generally exhibit high RAPSD in the low-frequency range. Thus, the RAPSD difference before and after applying a low-freq mask is relatively large, making random low-freq masking an effective way to supply sufficient learning signals, regardless of whether amplitude shifts degrade or remain unchanged, *i.e.*, for all types of domain shifts.

*2) High-frequency masking yields limited gains in* **SPA** *or even slightly hinder adaptation.* Images typically display low RAPSD at high frequencies, especially with further RAPSD degradation there, *e.g.*, blur corruptions. In these cases, masking high-freq components fails to provide sufficient learning signals, since the absolute RAPSD difference before and after applying a high-freq mask is relatively small. From Table 6, it can easily lead to trivial solutions if we only mask the high-freq amplitude. Moreover, masking high-freq components (which lowers RAPSD) runs counter

**Algorithm 1** PyTorch-style pseudocode of SPA.

```
# lf, hf: low-frequency, high-frequency
# a, m: \alpha for lf size, mask ratio, Eqn.(4)
# g: hyper-param \gamma for noise injection, Eqn.(5)

optimizer = SGD(model.learnable_params, lr, momentum)
predictions = []
for x in test_data_loader: # load data in online manner
    # structure-preserving random augmented views
    x1 = lf_aug(x, a, m) # lf amplitude mask, Eqn.(4)
    x2 = hf_aug(x, g) # hf noise injection, Eqn.(5)

    # forward-propagate
    y, y1, y2 = model(x), model(x1), model(x2)
    predictions.append(y)

    # active loss calculation, Eqn.(1)
    # use KL/L1 similarity for Cls./Reg. heads
    y = y.detach() # stop gradient
    loss = active_similarity_loss(y1, y)
    loss += active_similarity_loss(y2, y)

    loss.backward() # back-propagate
    optimizer.step() # parameter update
    optimizer.zero_grad()

return predictions
```

to the direction of domain shifts that raise RAPSD (*e.g.*, noise-based corruptions), thereby slightly hindering adaptation performance, as results on noisy corruptions in Table 6.

*3) Noise injection is an effective option for creating learning signals at high frequencies for* **SPA.** Though amplitude masking doesn't work at high frequencies, one can still augment the image to generate deteriorated views through noise injection to supply learning signals at high frequencies for self-bootstrapping learning. This is because noise injection, which is able to preserve the core geometry information of a given image, increases the information power/RAPSD at high frequencies, as in Figure 2 (a), thereby creating larger RAPSD differences before and after augmentation compared with high-freq amplitude masking.

Inspired by the above motivations, we derive our overall geometry-preserving augmentation from both low- and high-frequency perspectives, to supply sufficient learning signals for our self-bootstrapping TTA framework.

**Low-Frequency Amplitude Mask**  Let $\tau(\cdot)$ represent a shift (sort) operation that moves the low-frequency components to the center of the amplitude, and $\tau(\cdot)^{-1}$ be the inverse operation that returns the components to their original positions. Based on Eqn. (3), augmentation with a low-frequency amplitude mask becomes

$$\mathbf{v}_l = F^{-1}([\tau^{-1}\big(M(\alpha,m) \circ \tau(F^A(\mathbf{x}))\big), \ F^P(\mathbf{x})]), \quad (4)$$

where $M(\alpha,m)$ is a random 0-1 low-frequency mask with the same size as $\mathbf{x} \in \mathbb{R}^{h \times w}$. In $M(\alpha,m)$, the center area of size $\alpha h \times \alpha w$ is a random 0-1 matrix with a mask ratio $m$ and $\alpha$ is always set to 0.2, while the surrounding area is padded with 1. Other notations are the same as in Eqn. (3).

**High-Frequency Noise Injection**  We inject Gaussian noise into the original image to further supply learning signals at high frequencies. Note that although noise injection contrasts with the amplitude shift direction of some domain shifts like blur corruptions—where high-frequency RAPSD degrades—we still apply it in these cases and observe promising performance, *see* Table 6. This is because, for such corruptions, augmentations like high-frequency amplitude masking that align with the RAPSD degradation direction fail to provide effective learning signals. Instead, the availability of sufficient learning signals is more critical than maintaining consistency with the degradation direction. Consequently, noise injection remains an effective and viable option. Formally, the augmentation is given by

$$\mathbf{v}_h = (1 - \gamma) \cdot \mathbf{x} + \gamma \cdot \epsilon, \ \text{where} \ \epsilon \sim \mathcal{N}(0,1). \quad (5)$$

Here, $\mathcal{N}(0,1)$ is a multivariate standard normal distribution with the same size of image $\mathbf{x}$, $\gamma$ is a constant for injection.

## 4. Experimental Results

**Datasets and Models** For **classification**, we conduct experiments on four benchmarks, *i.e.*, ImageNet-C (Hendrycks & Dieterich, 2019) (corrupted images in 15 types of 4 main categories, with the most severe corruption level 5), ImageNet-R (artistic renditions of 200 ImageNet classes) (Hendrycks et al., 2021a), ImageNet-Adversarial (Hendrycks et al., 2021b) and ImageNet-Sketch (Wang et al., 2019). We use ViT-base (Dosovitskiy et al., 2021), trained on ImageNet by timm repository (Wightman, 2019), as the source model. For **3D monocular object detection**, we follow MonoTTA (Lin et al., 2024) to evaluate all methods on KITTI-C, constructed from a validation set of KITTI (Geiger et al., 2012) through the incorporation of 13 distinct types of data corruptions (Hendrycks & Dieterich, 2019). Each corruption has 3,769 images by following the original training and validation split of MonoFlex (Zhang et al., 2021). We use the model trained on KITTI by MonoFlex (Zhang et al., 2021) as the source model for TTA. For **segmentation**, we use the Segformer-B5 (Xie et al., 2021) model trained on Cityscape dataset (Cordts et al., 2016) as the source model and perform TTA on ACDC dataset (Sakaridis et al., 2021).

**Compared Methods** We compare SPA with: batch norm (BN) Adapt (Schneider et al., 2020); entropy-based: TENT (Wang et al., 2021), EATA (Niu et al., 2022a), SAR (Niu et al., 2023) and DeYO (Lee et al., 2024); ActMAD (Mirza et al., 2023) aligns feature statistics between target and source data; CoTTA (Wang et al., 2022) and DePT (Gao et al., 2023) adapt a given model via augmentation-based consistency maximization and a teacher-student learning scheme, ROID (Marsden et al., 2024) and CMF (Lee & Chang, 2024); MonoTTA (Lin

| Method | Noise | | | Blur | | | | Weather | | | | | Digital | | | Average |
|---|---|---|---|---|---|---|---|---|---|---|---|---|---|---|---|---|
| | Gauss. | Shot | Impul. | Defoc. | Glass | Motion | Zoom | Snow | Frost | Fog | Brit. | Contr. | Elastic | Pixel | JPEG | Acc. |
| Source | 56.8 | 56.8 | 57.5 | 46.9 | 35.6 | 53.1 | 44.8 | 62.2 | 62.5 | 65.7 | 77.7 | 32.6 | 46.0 | 67.0 | 67.6 | 55.5 |
| TENT | 60.3 | 61.6 | 61.8 | 59.2 | 56.5 | 63.5 | 59.2 | 54.3 | 64.5 | 2.3 | 79.1 | 67.4 | 61.5 | 72.5 | 70.6 | 59.6 |
| CoTTA | 63.6 | 63.8 | 64.1 | 55.5 | 51.1 | 63.6 | 55.5 | 70.0 | 69.4 | 71.5 | 78.5 | 9.7 | 64.5 | 73.4 | 71.2 | 61.7 |
| EATA | 62.2 | 63.4 | 63.4 | 60.5 | 61.2 | 66.0 | 63.5 | 70.3 | 68.4 | 73.1 | 79.8 | 67.0 | 69.7 | 75.2 | 73.4 | 67.8 |
| SAR | 59.2 | 60.5 | 60.7 | 57.5 | 55.6 | 61.8 | 57.6 | 65.9 | 63.5 | 69.1 | 78.7 | 45.7 | 62.4 | 71.9 | 70.3 | 62.7 |
| ActMAD | 61.3 | 62.8 | 63.2 | 55.9 | 55.7 | 62.7 | 61.7 | 70.8 | 68.8 | 73.5 | 80.8 | 62.3 | 67.8 | 74.8 | 73.0 | 66.3 |
| DeYO | 59.8 | 61.5 | 61.1 | 57.4 | 59.0 | 64.5 | 61.9 | 69.1 | 66.7 | 69.5 | 78.9 | 65.3 | 69.6 | 74.0 | 72.3 | 66.0 |
| ROID | 63.0 | 64.3 | 64.0 | 60.1 | 61.6 | 65.2 | 63.5 | 71.9 | 70.4 | 73.8 | 80.5 | 60.5 | 71.8 | 75.8 | 74.0 | 68.0 |
| CMF | 63.5 | 65.2 | 64.7 | 59.3 | 63.3 | 67.2 | 66.0 | 73.1 | 71.2 | 72.7 | 80.9 | 65.6 | 73.6 | 76.6 | 74.6 | 69.2 |
| SPA (ours) | 64.0 | 65.5 | 65.2 | 61.0 | 63.6 | 69.1 | 67.9 | 74.1 | 72.7 | 75.3 | 80.9 | 65.2 | 74.0 | 77.6 | 75.0 | $70.1_{\pm 0.1}$ |
| ✤ActMAD | 64.8 | 66.5 | 66.0 | 62.2 | 64.6 | 70.3 | 69.7 | 75.1 | 73.4 | 76.7 | 81.6 | 67.3 | 75.0 | 78.4 | 75.7 | $\underline{71.2_{\pm 0.2}}$ |
| ✤ActMAD+TENT | 65.2 | 67.0 | 66.4 | 63.7 | 65.7 | 70.9 | 70.4 | 75.2 | 73.5 | 77.2 | 81.7 | 67.8 | 75.5 | 78.4 | 76.0 | $\mathbf{71.6_{\pm 0.0}}$ |

*Table 1.* Comparisons with state-of-the-art methods on ImageNet-C (severity level 5) with ViT-Base regarding **Accuracy (%)**.

et al., 2024) and VDP (Gan et al., 2023) are TTA methods specified for 3D monocular detection and segmentation.

**Implementation Details** We set the mask ratio $m$ to 0.2 for all experiments. The noise factor $\gamma$ is set to 0.4 for classification and 0.1 for segmentation and 3D detection. Following CoTTA (Wang et al., 2022) and MonoTTA (Lin et al., 2024), we apply SGD on classification and 3D detection, and Adam on segmentation, using the learning rate of $10^{-2}/5\times10^{-3}/6\times10^{-5}$. We only update norm layers following TENT. More details of SPA and details of baseline methods are put in Appendix B.2. The cource code is availiable at https://github.com/mr-eggplant/SPA.

## 4.1. Image Classification

In this section, we validate our SPA on image classification. From results in Table 1, SPA outperforms all considered baselines consistently on all corruptions of ImageNet-C, highlighting its superiority. To be specific,

**1)** Compared to entropy-based methods such as TENT, EATA, SAR, and DeYO, SPA improves the state-of-the-art by 2.3% in average accuracy, achieving 70.1% compared to EATA's 67.8%. Additionally, SPA offers the flexibility to handle regression tasks (as shown in Table 3) while the entropy-based objectives can not; **2)** Compared to CoTTA, which also applies consistency learning between the original image and its augmented views, our SPA achieves significantly greater gains. This result highlights the effectiveness of our self-bootstrapping learning framework and demonstrates that our structure-preserving augmentations are able to provide richer learning signals within this framework; **3)** Compared to ActMAD, our method, SPA, does not require access to the source training data for calculating source statistics to achieve alignment, and thus is more general to be used in case of source data are unavailable. Despite this, SPA achieves higher performance than ActMAD, improving average accuracy from 66.3% to 70.1%; **4)** The learning objective of SPA is in parallel and not conflict with existing objectives like entropy minimization and feature

| Method | R | A | Sketch | Avg. Acc. (%,↑) |
|---|---|---|---|---|
| Source | 59.5 | 50.5 | 44.9 | 51.6 |
| TENT (Wang et al., 2021) | 63.9 | 52.8 | 49.1 | 55.3 |
| CoTTA (Wang et al., 2022) | 63.5 | 52.2 | 50.0 | 55.2 |
| EATA (Niu et al., 2022a) | 67.5 | 54.3 | 52.1 | 58.0 |
| ActMAD (Mirza et al., 2023) | 60.2 | 50.3 | 46.2 | 52.2 |
| DeYO (Lee et al., 2024) | $\underline{68.7}$ | $\underline{55.0}$ | 50.3 | 58.0 |
| SPA (ours) | 68.2 | **55.4** | $\underline{53.4}$ | $\underline{59.0}$ |
| ✤ EATA | **70.4** | $\underline{55.0}$ | **55.0** | **60.1** |

*Table 2.* Comparisons on ImageNet-R/A/Sketch with ViT-Base.

alignment, allowing it to be integrated with these approaches to enhance performance further, as demonstrated by SPA incorporating ActMAD and TENT. At last, from the results on ImageNet-R/A/Sketch in Table 2, our SPA also achieves the best performance, further suggesting our effectiveness.

## 4.2. 3D Monocular Object Detection

This section validates SPA on 3D monocular object detection (Zhang et al., 2021), a challenging task that involves detecting objects in single-camera images using 3D bounding boxes. This task comprises classification, to identify objects within each 3D bounding box, and regression, to predict the bounding box coordinates, dimensions, depths, and angles. We perform weak-to-strong self-bootstrapping learning to align the classification head (via KL loss) and align regression heads of bounding box coordinates, dims, and depths predictions (via L1 loss).

From Table 3, SPA achieves the best average $AP_{3D|R40}$ over 13 corruptions across different object difficulty levels (easy, moderate, hard) for each class, suggesting our effectiveness. Even compared to MonoTTA (Lin et al., 2024), the latest TTA method tailored for 3D monocular detection, SPA shows clear performance gains, with improvements such as 1.4% for Cars and 2.4% for Pedestrians. This largely benefits from the generality of SPA, which introduces TTA loss to regression heads in addition to classification heads, unlike previous methods, MonoTTA, TENT, and EATA, which focus solely on classification heads. Moreover, the 3D monocular detection task is highly imbalanced, with

| Method | Car, AP@0.7, 0.5, 0.5 | | | | Pedestrian, AP@0.5, 0.25, 0.25 | | | | Cyclist, AP@0.5, 0.25, 0.25 | | | |
|---|---|---|---|---|---|---|---|---|---|---|---|---|
| | Easy | Mod | Hard | Average | Easy | Mod | Hard | Average | Easy | Mod | Hard | Average |
| Source | 16.4 | 12.1 | 10.5 | 13.0 | 5.0 | 4.2 | 3.5 | 4.3 | 6.5 | 3.3 | 3.0 | 4.3 |
| BN Adapt (Schneider et al., 2020) | 33.2 | 23.0 | 19.2 | 25.1 | 9.7 | 8.1 | 6.7 | 8.2 | 12.9 | 6.6 | 6.0 | 8.5 |
| TENT (Wang et al., 2021) | 36.1 | 25.2 | 21.4 | 27.6 | 10.2 | 8.5 | 7.1 | 8.6 | 13.3 | 6.8 | 6.1 | 8.7 |
| EATA (Niu et al., 2022a) | 36.7 | 25.5 | 21.8 | 28.0 | 10.4 | 8.7 | 7.2 | 8.8 | 13.4 | 6.8 | 6.2 | 8.8 |
| ActMAD (Mirza et al., 2023) | 33.3 | 23.2 | 19.4 | 25.3 | 9.0 | 7.6 | 6.3 | 7.6 | 12.6 | 6.6 | 6.0 | 8.4 |
| MonoTTA (Lin et al., 2024) | 42.1 | 29.5 | 25.6 | 32.4 | 11.3 | 9.4 | 7.7 | 9.5 | 13.6 | 6.9 | 6.2 | 8.9 |
| SPA (ours) | 43.7 | 30.9 | 26.9 | $33.8_{\pm0.0}$ | 14.3 | 11.8 | 9.7 | $11.9_{\pm0.2}$ | 14.3 | 7.4 | 6.8 | $9.5_{\pm0.1}$ |
| ⊕ MonoTTA | 45.7 | 32.8 | 28.6 | $\mathbf{35.7}_{\pm0.1}$ | 14.9 | 12.3 | 10.1 | $\mathbf{12.4}_{\pm0.1}$ | 14.7 | 7.5 | 6.9 | $\mathbf{9.7}_{\pm0.2}$ |

*Table 3.* Comparisons on **3D monocular object detection** w.r.t. the average precision of 3D bounding boxes, denoted as $AP_{3D}|_{R40}(\%, \uparrow)$. The results are averaged over 13 corruptions of KITTI-C (*e.g.*, *Fog and Snow*, *see* Appendix B.1 for more details) with MonoFlex (Zhang et al., 2021) as the source model. The Intersection over Union (IoU) thresholds are set to 0.7, 0.5, 0.5 for Cars and 0.5, 0.25, 0.25 for Pedestrians and Cyclists, respectively.

| Time: $t \rightarrow$ | Round 1 | | | | | Round 2 | | | | | Round 3 | | | | | Round 1-3 |
|---|---|---|---|---|---|---|---|---|---|---|---|---|---|---|---|---|
| Method | Fog | Night | Rain | Snow | Avg. | Fog | Night | Rain | Snow | Avg. | Fog | Night | Rain | Snow | Avg. | Average |
| Source | 69.1 | 40.3 | 59.7 | 57.8 | 56.7 | 69.1 | 40.3 | 59.7 | 57.8 | 56.7 | 69.1 | 40.3 | 59.7 | 57.8 | 56.7 | 56.7 |
| TENT (Wang et al., 2021) | 69.1 | 40.2 | 60.0 | 57.3 | 56.7 | 68.4 | 39.1 | 60.0 | 56.4 | 56.0 | 67.6 | 37.9 | 59.7 | 55.3 | 55.1 | 55.9 |
| CoTTA (Wang et al., 2022) | 70.9 | 41.1 | 62.4 | 59.7 | 58.5 | 70.9 | 41.0 | 62.5 | 59.7 | 58.5 | 70.9 | 40.8 | 62.6 | 59.7 | 58.5 | 58.5 |
| DePT (Gao et al., 2023) | 71.0 | 40.8 | 58.2 | 56.8 | 56.7 | 68.2 | 40.0 | 55.4 | 53.7 | 54.3 | 66.4 | 38.0 | 47.3 | 47.2 | 49.7 | 53.6 |
| VDP (Gan et al., 2023) | 70.5 | 41.1 | 62.1 | 59.5 | 58.3 | 70.4 | 41.1 | 62.2 | 59.4 | 58.3 | 70.4 | 41.0 | 62.2 | 59.4 | 58.3 | 58.3 |
| SPA (ours) | 68.7 | 42.9 | 62.0 | 59.8 | 58.3 | 69.7 | 44.6 | 63.3 | 61.1 | 59.7 | 70.0 | 43.2 | 63.8 | 61.7 | 59.7 | $59.2_{\pm0.1}$ |
| ⊕ CoTTA | 71.2 | 42.7 | 65.2 | 62.1 | **60.3** | 72.5 | 43.1 | 66.0 | 62.2 | **61.0** | 72.5 | 42.9 | 66.0 | 62.1 | **60.9** | $\mathbf{60.7}_{\pm0.1}$ |

*Table 4.* Comparisons on **segmentation** under continual TTA. We report mIoU (%, ↑) on Cityscape-to-ACDC with Segformer-B5.

Pedestrian and Cyclist as minority objects. Methods that only target the classification head often over-optimize the majority class (*e.g.*, Cars) while neglecting minority ones. Thus, these classification-only methods yield very marginal gains on Pedestrian and Cyclist. In contrast, SPA is somehow free from this limitation, and it can mitigate this imbalanced issue, achieving higher gains on minority classes, *e.g.*, 2.4% higher $AP_{3D|R40}$ on Pedestrian over MonoTTA.

### 4.3. Image Segmentation

We compare SPA with prior TTA methods on image segmentation in a continual adaptation setting (Wang et al., 2022). In this setup, the target domains of ACDC (Sakaridis et al., 2021) progress through an ordered sequence of Fog→Night→Rain→Snow, repeated across 3 rounds, to simulate the environmental changes encountered in real-life driving scenarios. Table 4 shows that SPA alone outperforms CoTTA, which requires 29 augmentations per sample, while SPA achieves better results with only two augmentations, highlighting its effectiveness and efficiency. Furthermore, when combined with CoTTA, SPA improves the average mIoU from 58.5% to 60.7%, further demonstrating its superiority as a plug-and-play module.

### 4.4. Ablations

**Effects of Components in SPA** We ablate the effects of each component within SPA in Table 5. **First,** unlike previous SSL methods (Chen et al., 2020a; He et al., 2020) and augmentation consistency-based TTA methods (Zhang et al.,

| | ImageNet-C | KITTI-Fog $(AP_{3D|R40}, \%, \uparrow)$ | | |
|---|---|---|---|---|
| | Acc. (%, ↑) | Car | Pedestrian | Cyclist |
| No Adapt | 55.5 | 7.8 | 2.0 | 3.6 |
| Full SPA | **70.1** | **34.4** | **11.1** | **9.7** |
| *w/o weak-to-strong i.e., stop grad* | 4.9 | 24.0 | 7.3 | 5.4 |
| *w/o active selection, Eqn. (2)* | 69.4 | 28.0 | 9.9 | 7.7 |
| *w/o Low-freq amplitude mask* | 65.6 | 31.0 | 7.2 | 9.1 |
| *w/o High-freq noise injection* | 67.7 | 31.4 | 10.3 | 9.4 |

*Table 5.* Ablation on effects of components in SPA. We use ViT-Base/MonoFlex as the source model on ImageNet-C/KITTI-Fog.

2022) that maximize prediction or feature consistency between two augmented views, SPA adopts a weak-to-strong self-bootstrapping learning paradigm, which learns from a weak (augmented) view to a strong (original) view. This unidirectional learning plays a crucial role in our TTA approach, as it provides more reliable learning signals, which are essential in TTA contexts. Without this weak-to-strong mechanism, bidirectional consistency learning results in drastic performance degradation. **Second,** The activation selection strategy in Eqn. (2) also helps to filter out partially unreliable supervisions, based on the premise that predictions from the original image are more reliable than those from the weaker augmented views, and thus results in promising performance gains. **Third,** both our low-frequency amplitude mask and high-frequency noise injection augmentation strategies are effective individually, yet achieve the best performance when applied together.

**Effects of Different Geometric-Preserving Augmentations in SPA** The core idea in SPA is to devise image geometric structure-preserving augmentations to boost its applicability for fine-grained tasks like object detection

| | Aug. Choice | Noise | | | Blur | | | | Weather | | | | Digital | | | | Average |
|---|---|---|---|---|---|---|---|---|---|---|---|---|---|---|---|---|---|
| | | Gauss. | Shot | Impul. | Defoc. | Glass | Motion | Zoom | Snow | Frost | Fog | Brit. | Contr. | Elastic | Pixel | JPEG | Acc. |
| a) | *Greystyle* | 59.7 | 59.8 | 61.4 | 58.7 | 59.0 | 65.5 | 63.6 | 70.3 | 69.4 | 72.5 | 76.5 | 0.4 | 71.0 | 75.5 | 71.4 | 62.3 |
| b) | *Brightness* | 38.4 | 60.5 | 60.8 | 9.7 | 55.5 | 35.5 | 60.9 | 69.2 | 70.3 | 4.9 | 79.7 | 0.8 | 66.0 | 74.3 | 70.7 | 50.5 |
| c) | *Contrast* | 34.9 | 61.3 | 61.6 | 45.0 | 58.2 | 65.0 | 61.5 | 68.1 | 70.0 | 72.2 | 78.7 | 0.4 | 66.9 | 74.5 | 70.9 | 59.3 |
| d) | *Gaussian Blur* | 3.2 | 8.7 | 2.9 | 20.9 | 1.7 | 30.6 | 30.4 | 57.9 | 41.8 | 73.9 | 80.2 | 6.5 | 14.6 | 2.0 | 6.6 | 25.4 |
| | a) + b) + c) | 59.8 | 60.5 | 61.3 | 60.0 | 60.5 | 66.6 | 64.1 | 70.5 | 69.2 | 73.7 | 76.5 | 0.4 | 71.4 | 75.8 | 71.7 | 62.8 |
| e) | *Freq Mask* | 60.8 | 62.5 | 61.7 | 57.8 | 59.4 | 66.5 | 63.9 | 69.9 | 68.0 | 74.8 | 79.6 | 65.6 | 70.7 | 75.3 | 72.7 | 67.3 |
| f) | *High-Freq Mask* | 4.3 | 5.9 | 5.4 | 1.4 | 6.6 | 10.5 | 16.0 | 63.7 | 21.3 | 22.9 | 73.4 | 2.3 | 11.0 | 2.5 | 4.7 | 16.8 |
| g) | *Low-Freq Mask* | 62.6 | 64.3 | 63.4 | 57.6 | 59.2 | 66.9 | 64.4 | 70.6 | 69.0 | 74.8 | 79.6 | 65.7 | 71.0 | 75.9 | 72.6 | 67.8 |
| h) | *Noise Injection* | 63.6 | 65.5 | 65.2 | 57.5 | 61.2 | 66.5 | 63.9 | 73.6 | 72.1 | 68.4 | 81.0 | 21.6 | 71.7 | 77.1 | 74.8 | 65.6 |
| | g) + h) (ours) | 64.0 | 65.5 | 65.2 | 61.0 | 63.6 | 69.1 | 67.9 | 74.1 | 72.7 | 75.3 | 80.9 | 65.2 | 74.0 | 77.6 | 75.0 | **70.1** |

*Table 6.* Effects of different image geometric structure-preserving augmentation choices under our self-bootstrapping learning framework. We report **Accuracy (%)** on ImageNet-C (severity level 5) with ViT-Base.

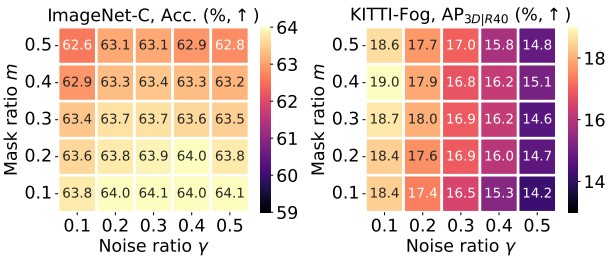

*Figure 3.* Sensitivity of amplitude mask ratio $m$ in Eqn. (4) and noise injection ratio $\gamma$ in Eqn. (5). We use ViT-Base for ImageNet-C (Gaussian Noise) and MonoFlex for KITTI-Fog. The source model Acc./AP on ImageNet-C/KITTI-Fog is 55.5%/4.5%.

| Method | ImageNet-C Acc. (%, ↑) | KITTI-Fog (AP$_{3D|R40}$, %, ↑) | | |
|---|---|---|---|---|
| | | Car | Pedestrian | Cyclist |
| No Adapt | 55.5 | 7.8 | 2.0 | 3.6 |
| BN Adapt (Schneider et al., 2020) | n/a | 23.3 | 8.6 | 9.7 |
| *Self-bootstrapping learning of SPA using augmentations of:* | | | | |
| Eqns. (4) and (5) (ours) | **70.1** | **34.4** | **11.1** | **9.7** |
| Random image mask | 64.9 | 9.7 | 4.3 | 2.0 |
| AugMix | 63.5 | 26.8 | 6.2 | 6.0 |
| SimCLR augmentations | 68.1 | 25.0 | 5.7 | 6.6 |
| MoCo augmentations | 64.0 | 27.7 | 6.5 | 6.7 |

*Table 7.* Comparisons with SPA using conventional augmentation strategies that do not preserve geometric structure. We use ViT-Base/MonoFlex as the source model on ImageNet-C/KITTI-Fog.

and segmentation. Here, we compare our augmentation strategy with existing geometric-preserving ones, including *grayscale, brightness, contrast (ColorJitter)*, and *Gaussian blur*. However, as in Table 6, none of them, individually or combined, effectively provide TTA with rich learning signals. These augmentations are also often sensitive to the corruption type and struggle to perform stably across all corruptions, leading to limited overall performance. Moreover, as discussed in Sect. 3.2, SPA masks only the low-frequency amplitude while keeping the high-frequency components unchanged, since the high-frequency range exhibits low RAPSD and masking there provide limited learning signals (whereas low frequencies are quite the opposite). To verify this, rows f) and g) in Table 6 show that solely masking high frequencies tends to yield trivial solutions, *i.e.*, model collapse, while masking at low frequencies perform stably.

**Parameter Sensitivity** We evaluate SPA with different amplitude mask ratio $m$ (in Eqn. (4)) and noise ratio $\gamma$ (in Eqn. (5)) selected from {0.1, 0.2, 0.3, 0.4, 0.5}. From Figure 3, SPA works well across a wide range of $m \leq 0.5$ and $\gamma \leq 0.5$ for image classification, showing its insensitivity. However, for 3D monocular detection, although $m$ performs well within a broad range from 0.1 to 0.5, the optimal range of noise ratio $\gamma$ is narrower than that in image classification, *i.e.*, $\gamma \leq 0.2$. This difference arises because, in ImageNet-C, the task is at the image level and does not

strictly require content invariance, allowing for a higher $\gamma$ to provide richer learning signals. In contrast, 3D monocular detection involves dense predictions where high noise levels could significantly disrupt the original image content, making it challenging for our self-bootstrapping learning.

**Comparison with Self-Bootstrapping Learning using Augmentations of Image Mask (He et al., 2022; Gandelsman et al., 2022b), AugMix (Hendrycks et al., 2020), MoCo (He et al., 2020) and SimCLR (Chen et al., 2020a)** We compare our proposed augmentation strategy with prior augmentations used in SSL and consistency-based TTA methods under our self-bootstrapping TTA framework in Table 7. Results show that while prior augmentations achieve considerable performance on image-level classification (ImageNet-C), they fall short for 3D monocular detection, where they often perform worse than BN Adapt. This arises because these augmentations like random cropping&resizing disrupt the image's geometric structure, making them unsuitable for finer dense prediction tasks. In contrast, our augmentations are geometric-preserving and supply rich signals for weak-to-strong self-bootstrapping learning, achieving improved adaptation performance.

### 4.5. Further Results and Discussions

**Comparisons with Methods Beyond Fully TTA** Our SPA approach can adapt any pre-trained model off-the-shelf without any other requirements, which falls into the category of

| Model + Method | Gauss. | Fog | Pixel | Snow | Contr. | Average |
|---|---|---|---|---|---|---|
| Customized ViT-L/16 classifier | 17.1 | 38.7 | 47.1 | 35.6 | 6.9 | 29.1 |
| ✚ TTT-MAE | 37.9 | 51.1 | 65.7 | 56.5 | 10.0 | 44.2 |
| ViT-B/32 | 39.5 | 35.9 | 55.0 | 30.0 | 31.5 | 38.4 |
| ✚ Diffusion-TTA | 46.5 | 56.2 | 64.7 | 50.4 | 33.6 | 50.3 |
| ✚ SPA (Ours) | 49.4 | 56.9 | 66.7 | 49.9 | 51.5 | 54.9 |

*Table 8.* Comparison with TTT-MAE and Diffusion-TTA on ImageNet-C regarding accuracy (%).

| Method | Source | TPT | C-TPT | ETA | SAR | DeYO | SPA | SPA+ETA |
|---|---|---|---|---|---|---|---|---|
| Acc (%) | 74.0 | 77.1 | 76.0 | 76.9 | 75.6 | 76.6 | 77.2 | 78.2 |

*Table 9.* Comparisons with baselines on CLIP-ViT-B for OOD robustness. We report results on ImageNet-R.

| Method | DTD | UCF101 | Aircraft | Avg. Acc. (%) |
|---|---|---|---|---|
| Source | 44.3 | 65.1 | 23.8 | 44.4 |
| *VLM TTA methods:* | | | | |
| TPT (Shu et al., 2022) | 46.7 | 67.3 | 23.4 | 45.8 |
| C-TPT (Yoon et al., 2024) | 46.0 | 65.7 | 24.0 | 45.2 |
| *Vision TTA methods:* | | | | |
| Tent | 45.2 | 66.0 | 23.4 | 44.9 |
| ETA | 44.7 | 66.1 | 23.7 | 44.8 |
| SAR | 44.6 | 66.5 | 23.4 | 44.8 |
| DeYO | 44.2 | 66.0 | 22.7 | 44.3 |
| SPA (ours) | 45.4 | 66.2 | 23.6 | 45.1 |

*Table 10.* Results on CLIP-ViT-B for cross-dataset generalization.

fully TTA. Here, we further compare SPA with methods beyond fully TTA, *i.e.*, TTT-MAE (Gandelsman et al., 2022a) that requires modifying the original model training process and Diffusion-TTA (Prabhudesai et al., 2023) that relies on an additional pre-trained generative diffusion model and updates both the diffusion model and original discriminative model at test time. As shown in Table 8, our SPA, as a fully TTA method, still achieves superior performance, further demonstrating our effectiveness.

**Effectiveness on Vision-Language Model (VLM) for both OOD and Cross-Dataset Generalization** We freeze the text encoder of the CLIP (Radford et al., 2021) model and treat it as a fixed classifier, and then apply SPA to the image encoder. SPA primarily focuses on addressing the OOD issue in the visual modality through our deterioration-driven self-bootstrapping learning, by improving the visual feature representation. Therefore, as in Table 9, SPA achieves superior performance on OOD benchmarks.

However, for commonly used cross-dataset benchmarks (like DTD (Cimpoi et al., 2014), UCF101 (Soomro et al., 2012) and Aircraft (Maji et al., 2013)) on the CLIP model, the image distributions are often relatively more stable compared to OOD datasets, and the image encoder already provides more semantical representations. The performance bottleneck instead often lies in the text modality (*i.e.*, the text encoder)—specifically, the quality of the classifier formed from text embeddings. This is also supported by 1) prior VLM TTA methods, such as TPT (Shu et al., 2022) and C-TPT (Yoon et al., 2024), which focus on adapting the text branch to improve cross-dataset generalization, and 2) prior visual modality-focused TTA methods (like Tent and ETA) achieve limited gains in this cross-dataset scenario.

Therefore, the performance gain of our SPA method on cross-dataset scenarios with CLIP is not as competitive as its improvement on OOD scenarios—but it still provides benefits (*see* Table 10)—as we do not adapt the text branch. Extending our method to also adapt the text branch to further boost performance in cross-dataset scenarios is an interesting and promising direction. We leave this for future work.

## 5. Conclusion

In this paper, we aim to develop a new versatile fully TTA approach to support various tasks—classification or regression across image-, object-, and pixel-level predictions. To this end, we re-establish a consistency-based TTA framework with active weak (augmented image)-to-strong (original image) supervisions, *termed* SPA. In SPA, by analyzing how domain shifts manifest in the Fourier frequency domain, we devise two Fourier-based augmentations: low-frequency amplitude masking and high-frequency noise injection. These augmentations preserve the geometric structure of images (*e.g.*, object locations and sizes), making them applicable for consistency learning in fine-grained dense tasks while also supplying rich learning signals for TTA. Extensive experiments on classification, 3D monocular object detection, and segmentation verify the generality and superiority of SPA, both as a standalone approach and as a plug-and-play module to enhance existing methods.

## Impact Statement

This paper presents work whose goal is to advance the field of Machine Learning, especially for Test-Time Machine Learning. There are many potential societal consequences of our work, none of which we feel must be specifically highlighted here.

## Acknowledgment

This research was supported, in part, by the Joint WeBank-NTU Research Institute on Fintech, Nanyang Technological University, Singapore, and Ministry of Education, Singapore, under its Academic Research Fund Tier 1.

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

# Self-Bootstrapping for Versatile Test-Time Adaptation
## Supplementary Materials

## A. Related Work

We categorize related **Test-Time Adaptation (TTA)** works based on their learning objectives and discuss their generality across model architectures and tasks. Following that, we relate our method with data augmentation techniques.

**Learning Objective-Free TTA** mainly relies on adapting batch norm statistics (Nado et al., 2020; Schneider et al., 2020; Gong et al., 2022). Building on this, various methods have been developed to extend its applicability to diverse test scenarios, *e.g.*, label shifts (Niu et al., 2023; Gong et al., 2022) and single sample (Khurana et al., 2021; Niu et al., 2023), using methods like data augmentation (Khurana et al., 2021), statistics mix-up (Hu et al., 2021; Lim et al., 2023), re-normalization (Zhao et al., 2023), *etc.* These methods are general for various tasks (*e.g.*, image-/object-level) but are limited to batch norm-equipped models.

**Learning-Based TTA** (Sun et al., 2020; Wang et al., 2021; Liang et al., 2023; Niu et al., 2022b; 2024; Deng et al., 2024; Chen et al., 2024a; Zhang et al., 2025) explicitly learns from test data by updating model parameters using self- or unsupervised learning, achieving much better gains. Unlike learning-free TTA, in learning-based TTA, the designed objectives typically support various models, *e.g.*, CNN or ViT, but may be restrictively applicable to different tasks:

• ***Entropy / Self-Learning***-based methods (Wang et al., 2021; Mummadi et al., 2021; Goyal et al., 2022; Chen et al., 2024b) are among the most popular fully TTA approaches. These methods, which optimize prediction entropy or cross-entropy loss using pseudo-labels, have become foundational techniques in the TTA community. Building on them, several methods have been further developed: EATA (Niu et al., 2022a; Tan et al., 2024) and SAR (Niu et al., 2023) introduce selective entropy minimization strategies for improved efficiency and sharpness-aware optimization for stable TTA in the wild; SLR (Mummadi et al., 2021) and Conjugate PL (Goyal et al., 2022) design advanced loss functions to better utilize pseudo-labels in TTA, *etc.*

• ***Prototype-Based TTA*** methods (Iwasawa & Matsuo, 2021; Choi et al., 2022) enhance TTA performance by maintaining class-specific prototypes and making predictions based on feature similarity with these prototypes.

However, all these methods rely on predicted class probabilities, making them unsuitable for regression tasks.

• ***Contrastive-Based*** TTA (Zhang et al., 2022; Liu et al., 2021; Shu et al., 2022; Chen et al., 2022) is another popular and promising direction. It is mainly based on self-supervised learning methods that seek to learn robust representations by contrasting different augmentation views of the same image to enforce consistency, such as MoCo (He et al., 2020; Chen et al., 2020b), SimCLR (Chen et al., 2020a), BYOL (Grill et al., 2020), and DINO (Caron et al., 2021). Building on this, many test-time adaptation (TTA) methods incorporate contrastive learning objectives to adapt models to out-of-distribution data during testing, showing promising performance. TTT++ (Liu et al., 2021) exploits the SimCLR (Chen et al., 2020a) loss, while Ada-Contrast (Chen et al., 2022) incorporates MoCo (He et al., 2020)'s momentum update scheme with the InfoNCE loss, which conduct contrastive learning at the feature level. MT3 (Bartler et al., 2022) explores BYOL (Grill et al., 2020) loss for TTA, applying contrastive learning directly on model predictions. However, as a TTT-series method, MT3 still requires modifications to the model training process. In contrast, MEMO (Zhang et al., 2022) and TPT (Shu et al., 2022) extend this prediction-level consistency learning into a fully TTA approach by maximizing prediction consistency across various augmented views.

Nevertheless, the success of contrastive learning (He et al., 2020; Chen et al., 2020a; Grill et al., 2020; Caron et al., 2021) and their integration in TTA (Zhang et al., 2022; Shu et al., 2022; Chen et al., 2022; Bartler et al., 2022) stems from a crucial motivation: they depend on heavy and strong augmentations to supply adequate learning signals across different views. Their commonly used strong augmentations, *e.g.*, random cropping, resizing, and masking, often disrupt the overall content and geometric structure of the image, making them work well for image-level tasks, but infeasible for object-/pixel-level TTA tasks. In this work, we aim to develop a new versatile fully TTA method to enable model- and task-agnostic applications, by augmenting an image in the Fourier domain for self-bootstrapping.

**Data Augmentation** (Shorten & Khoshgoftaar, 2019) aims to increase the diversity of training data, *i.e.*, enlarging training data distribution, by applying transformations such as rotation, flipping, and cropping, helping models generalize better and reduce overfitting. As a widely used technique in deep learning, data augmentation has evolved significantly over time. Starting from the simple, manually designed

transformations, it has progressed to more advanced approaches like RandAugment (Cubuk et al., 2020), AutoAugment (Cubuk et al., 2019), FastAutoAugment (Lim et al., 2019), Mixup (Zhang, 2017), AugMix (Hendrycks et al., 2020) and *etc.* These methods have demonstrated remarkable effectiveness in enhancing generalization at training time in supervised manner, and also have been adopted by self-supervised (He et al., 2020; Chen et al., 2020b;a; Grill et al., 2020; Caron et al., 2021) methods and contrastive TTA methods (Chen et al., 2022; Zhang et al., 2022; Bartler et al., 2022) to generate diverse augmentation views for effective learning, as we depicted above.

In the work, we develop a new versatile fully TTA method, *termed* SPA, to enable model- and task-agnostic applications. SPA is also built on data augmentation. It augments/deteriorates an image in the Fourier domain to generate a weak view for self-bootstrapping learning. Here, we would like to point out that although recently there are some works introducing Fourier-based augmentations (Xu et al., 2023; Yang & Soatto, 2020; Kalibhat et al., 2023), they target training-time generalization, unsupervised domain adaptation and self-supervised learning, exploring techniques like amplitude swap/mixup, and phase shift. And these methods still overlook and disrupt the geometric structure of images after augmentation, or rely on paired source and target data, making them incompatible with our context of a versatile fully TTA framework. In this paper, we analyze how common distribution shifts manifest in the Fourier domain and, based on our findings, design specific augmentation strategies for low and high frequencies, respectively. Our augmentations aim to preserve the core geometric structure of the image while providing as much as possible learning signals for effective and versatile TTA.

## B. More Implementation Details

### B.1. More Details on Datasets

We conduct experiments on six datasets to evaluate the OOD generalization. Specifically, 1) for classification: we use ImageNet-C (Recht et al., 2019), ImageNet-R (Hendrycks et al., 2021a), ImageNet-A (Hendrycks et al., 2021b), and ImageNet-Sketch (Wang et al., 2019); 2) for 3D monocular object detection: we use KITTI-C per MonoTTA (Lin et al., 2024); and 3) for segmentation: we use ACDC (Sakaridis et al., 2021) per CoTTA (Wang et al., 2022); encompassing **35 distribution shifts** in total, as shown in Figure 4.

**ImageNet-C** consists of various versions of corruption applied to 50,000 validation images from ImageNet. The dataset encompasses 15 distinct corruption types of 4 main groups, including noise, blur, weather, and digital. Each corruption is characterized by 5 different levels of severity. We specifically utilize severity level 5 for all evaluations.

**ImageNet-R** contains 30,000 images featuring artistic renditions of 200 ImageNet classes. These images are mainly sourced from Flickr and filtered by Amazon MTurk.

**ImageNet-A** comprises 7,500 images covering 200 ImageNet classes. These images are naturally existing samples that lead to a notable degradation in classifier performance.

**ImageNet-Sketch** consists of 50,899 images represented as black and white sketches, covering 1000 ImageNet classes.

**KITTI-C** consists of various versions of corruption applied to 3,769 validation images from KITTI (Geiger et al., 2012). It encompasses 13 corruption types of 4 main groups, including *Gaussian noise, shot noise, impulse noise, defocus blur, glass blur, motion blur, snow, frost, fog, brightness, contrast, pixelation,* and *saturation.* Each corruption is characterized by 5 different levels of severity. We utilize severity level 1 for all evaluations per MonoTTA (Lin et al., 2024).

**ACDC** contains four categories of images collected in adverse conditions, including *fog, night, rain,* and *snow.* Following CoTTA (Wang et al., 2022), we use 400 unlabeled images from each adverse condition for continuous TTA.

### B.2. More Evaluation Protocols

We use the ViT-Base (Dosovitskiy et al., 2021) model trained on ImageNet by timm (Wightman, 2019) as the source model for classification, the MonoFlex (Zhang et al., 2021) model trained on KITTI (Geiger et al., 2012) for 3D monocular detection, and the Segformer-B5 (Xie et al., 2021) model trained on CityScapes (Cordts et al., 2016) for semantic segmentation. We introduce the implementation details of the involved methods below.

**SPA (ours)** For **classification**, we set the mask ratio $m$ to 0.2 and the noise factor $\gamma$ to 0.4. We insert a new learnable projector before the final predictions of the augmented (weak) views following BYOL (Grill et al., 2020). This projector is initialized as identity mappings and updated via SGD optimizer with a learning rate of 0.05 and a momentum of 0.9. The affine parameters of the norm layers are also updated via SGD, using a learning rate of 0.01 and a momentum of 0.9. When integrated with ActMAD (Mirza et al., 2023) and Tent (Wang et al., 2021), the learning rate of the norm layers is set to 0.005 following ActMAD. For **3D monocular object detection**, we set the mask ratio $m$ to 0.2 and the noise factor $\gamma$ to 0.1. We align the classification head via KL loss and regression heads of bounding box coordinates, dims, and depths prediction via L1 loss, where we apply a rescaling factor of 0.01 on regression losses for balancing. We also adopt a threshold of 0.4 on $\hat{f}(\mathbf{x}; \theta)$ to filter unreliable predictions inspired by MonoTTA (Lin et al., 2024). The norm layers are updated using SGD, with a learning rate of 0.005 and a momentum of 0.9. When integrated with MonoTTA, the loss com-

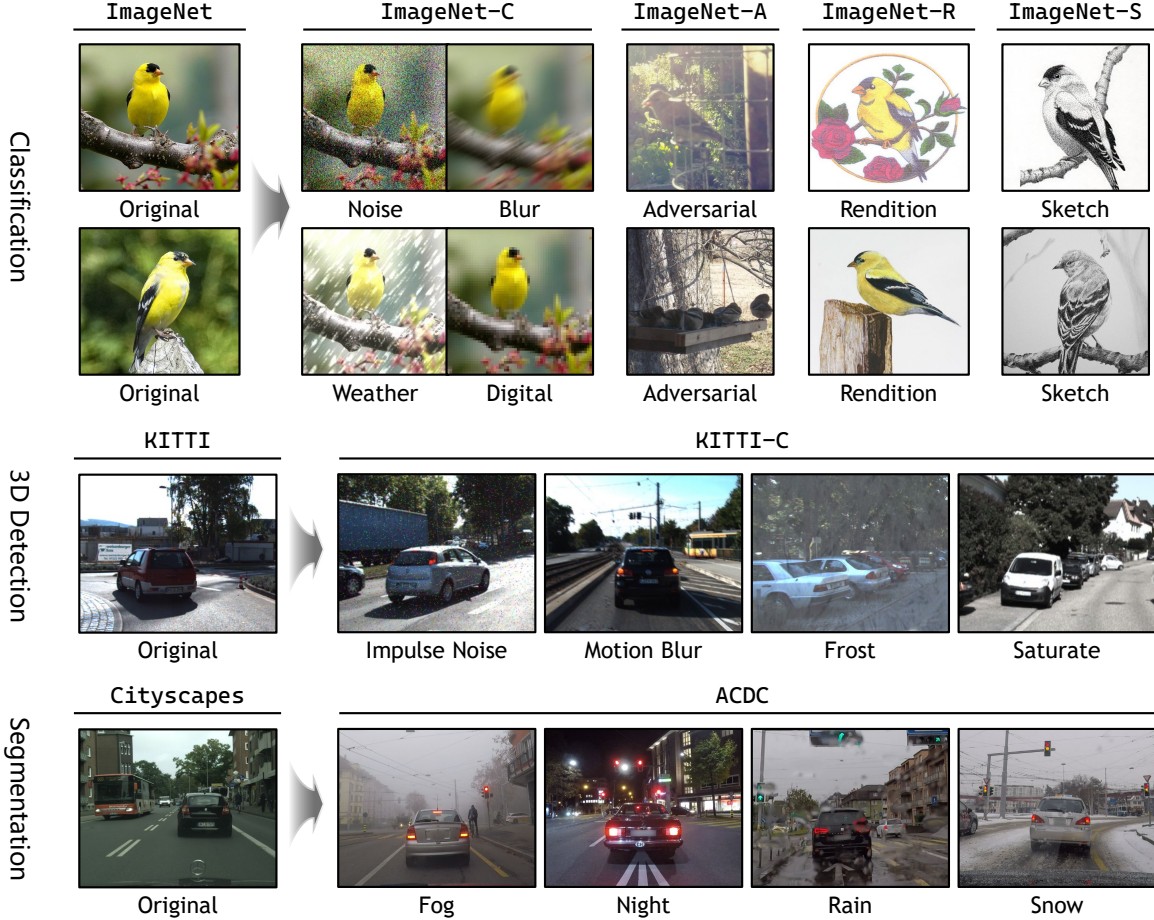

*Figure 4.* Visualizations of partial images in ImageNet, ImageNet-C/A/R/Sketch, KITTI, KITTI-C, Cityscapes and ACDC.

ponents from SPA are further rescaled by a factor of 0.5. For **semantic segmentation**, when incorporated with CoTTA (Wang et al., 2022), we set the mask ratio $m$ to 0.2 and the noise factor $\gamma$ to 0.1, where our augmentation strategies are applied on the student model. We adopt the Adam optimizer with the learning rate of $6 \times 10^{-5}$ to optimize all the parameters in Segformer-B5. When SPA works as a standalone method, we further apply a confidence threshold of 0.2 on the original (strong) views to select reliable predictions inspired by CoTTA (Wang et al., 2022), and adopt the Adam optimizer to update the norm layers with a learning rate of $1 \times 10^{-4}$.

We make noise injection learnable for image classification tasks by dividing the images into patches, with a patch size of 16, and learn patch-specific noise factor $\{\gamma_i\}_{i=1}^{P}$, where $P$ is the number of patches. $\boldsymbol{\gamma}$ is updated to maximize the test-time objective via SGD with a learning rate of 1 and a momentum of 0.9, under the constraint $\text{mean}(\boldsymbol{\gamma}) = 0.4$.

**TENT**[1] (**Wang et al., 2021**) We follow all hyper-parameters

that are set in Tent unless it does not provide. For classification, we use the SGD optimizer, with a momentum of 0.9 and a learning rate of 0.001. For 3D monocular detection, we use the SGD optimizer per MonoTTA (Lin et al., 2024), with a momentum of 0.9 and a learning rate of 0.0005, and entropy loss is applied on the classification head. For segmentation, we use the Adam optimizer per CoTTA (Wang et al., 2022) with a learning rate of 0.00006/8. Trainable parameters are the parameters of the norm layers.

**SAR**[2] (**Niu et al., 2023**) We follow all hyper-parameters that are set in SAR unless it does not provide. Specifically, we use SGD as the update rule, with a momentum of 0.9, batch size of 64, and a learning rate of 0.001. The threshold $E_0$ is set to $0.4 \times \ln C$, where $C$ is the number of classes. The trainable parameters are the affine parameters of the layer normalization layers from blocks 1 to blocks 8 in ViT-Base (Dosovitskiy et al., 2021).

**EATA**[3] (**Niu et al., 2022a**) We follow all hyper-parameters

---

[1] https://github.com/DequanWang/tent

[2] https://github.com/mr-eggplant/SAR
[3] https://github.com/mr-eggplant/EATA

| Method | Noise | | | Blur | | | | Weather | | | | | Digital | | | Average |
|--------|-------|------|-------|-------|-------|--------|------|------|-------|------|-------|-------|---------|-------|------|---------|
| | Gauss. | Shot | Impul. | Defoc. | Glass | Motion | Zoom | Snow | Frost | Fog | Brit. | Contr. | Elastic | Pixel | JPEG | Acc. |
| Source | 56.8 | 56.8 | 57.5 | 46.9 | 35.6 | 53.1 | 44.8 | 62.2 | 62.5 | 65.7 | 77.7 | 32.6 | 46.0 | 67.0 | 67.6 | 55.5 |
| TENT | 60.3 | 61.6 | 61.8 | 59.2 | 56.5 | 63.5 | 59.2 | 54.3 | 64.5 | 2.3 | 79.1 | 67.4 | 61.5 | 72.5 | 70.6 | 59.6 |
| EATA | 62.2 | 63.4 | 63.4 | 60.5 | 61.2 | 66.0 | 63.5 | 70.3 | 68.4 | 73.1 | 79.8 | 67.0 | 69.7 | 75.2 | 73.4 | 67.8 |
| SPA | 64.0 | 65.5 | 65.2 | 61.0 | 63.6 | 69.1 | 67.9 | 74.1 | 72.7 | 75.3 | 80.9 | 65.2 | 74.0 | 77.6 | 75.0 | 70.1 |
| SPA-I | 62.7 | 64.3 | 64.2 | 58.9 | 61.9 | 67.7 | 65.9 | 72.6 | 71.7 | 75.1 | 80.5 | 66.3 | 71.9 | 76.8 | 73.6 | 69.0 |

*Table 11.* Effects of combined *vs.* separate augmentation strategy in SPA on ImageNet-C (severity level 5) with ViT-Base w.r.t. **Accuracy (%)**. **SPA-I** applies low-frequency amplitude mask (Eqn. (4)) and high-frequency noise injection (Eqn. (5)) in a single image simultaneously, obtaining one augmented image. While SPA augments an image using Eqn. (4) and Eqn. (5) separately, generating two augmented images for test-time self-bootstrapping learning.

that are set in EATA unless it does not provide. Specifically, for image classification, we use SGD as the update rule, with a momentum of 0.9 and a learning rate of 0.001. The entropy threshold $E_0$ is set to $0.4 \times \ln C$, where $C$ is the number of task classes. We use 2,000 samples to estimate the importance of each parameter. The trainable parameters are all affine parameters of layer normalization layers in ViT-Base (Dosovitskiy et al., 2021). For 3d monocular detection, we follow the hyper-parameters specified by MonoTTA (Lin et al., 2024). In particular, we optimize the affine parameters of the batch norm layers using SGD, with a momentum of 0.9 and a learning rate of 0.0005. The entropy threshold $E_0$ is set to $\ln C/2 - 1$ and we also apply a confidence threshold of 0.2 to filter out potentially unreliable predictions.

**DeYO**[4] **(Lee et al., 2024)** We follow all hyper-parameters that are set in DeYO unless it does not provide. Specifically, we use SGD as the update rule, with a momentum of 0.9 and a learning rate of 0.001. The entropy threshold $E_0$ is set to $0.4 \times \ln C$ and the entropy factor $\tau_{\text{Ent}}$ is set to $0.5 \times \ln C$, where $C$ is the number of task classes. The Pseudo-Label Probability Difference (PLPD) threshold $\tau_{\text{PLPD}}$ is set to 0.2. Trainable parameters are the affine parameters of the layer normalization layers from blocks 1 to blocks 8 in ViT-Base (Dosovitskiy et al., 2021).

**ActMAD**[5] **(Mirza et al., 2023)** The implementations of ActMAD on ViT-Base (Dosovitskiy et al., 2021) for classification and MonoFlex (Zhang et al., 2021) for 3D monocular detection are inspired by FOA. For classification, we calculate the source training statistics with the validation set of ImageNet and align the test statistics per batch, using the SGD optimizer with a learning rate of 0.005 and a momentum of 0.9. For 3D monocular detection, we calculate the source training statistics with 15,168 samples from the validation set of KIITI and adopt the SGD optimizer with a learning rate of 0.0005 and a momentum of 0.9. Trainable parameters are affine parameters in norm layers.

**CoTTA**[6] We follow all hyper-parameters that are set in CoTTA unless it does not provide. For classification, we use SGD as the update rule, with a momentum of 0.9, and a batch size of 64. We consistently set the learning rate to 0.001 and the augmentation threshold $p_{th}$ to 0.1 given the optimal accuracy observed in Table 1. For images below the threshold, we conduct 32 augmentations including color jitter, random affine, Gaussian blur, random horizontal flip, and Gaussian noise. For segmentation, the threshold $p_{th}$ is set to 0.69. We apply a range of image resolution scale factors [0.5, 0.75, 1.0, 1.25, 1.5, 1.75, 2.0] and horizontal flip as the augmentation of teacher model input. We use Adam as the update rule with a learning rate of 0.00006/8. In both cases, the trainable parameters are all the parameters in the student model, and the teacher model is updated via the exponential moving average with a moving factor of 0.999. The restoration probability is set to 0.01.

**MonoTTA**[7] **(Lin et al., 2024)** We follow all hyper-parameters that are set in MonoTTA. Specifically, the initial object detection threshold $\gamma$ is set to 1, the decay coefficient $\beta$ for threshold updating is set to 0.9, and the threshold $\eta$ for low-score object filtering is set to 0.05. The affine parameters of batch norm layers are updated via SGD, with a learning rate of 0.0005 and a momentum of 0.9. We also follow the hyper-parameters of other baseline methods, including TENT, EATA, and ActMAD, as specified by MonoTTA for the 3D monocular object detection task.

## C. More Discussions and Results

**Effects of Combined *vs.* Separate Augmentations in SPA** In SPA, we augment a given image using low-frequency amplitude mask in Eqn. (4) and high-frequency noise injection in Eqn. (5) to generate two distinct augmented views for weak-to-strong self-bootstrapping learning. This separates the learning process for high and low frequencies and helps the model better learn low- or high-frequency preserving features, aiming to maximize the efficiency of utilizing the

---

[4] https://github.com/Jhyun17/DeYO
[5] https://github.com/jmiemirza/actmad

[6] https://github.com/qinenergy/cotta
[7] https://github.com/Hongbin98/MonoTTA

| | KITTI-Fog ($AP_{3D|R40}$, %, ↑) | | | |
|---|---|---|---|---|
| Method | Car | Pedestrian | Cyclist | Avg. |
| No Adapt | 7.8 | 2.0 | 3.6 | 4.5 |
| BN Adapt (Schneider et al., 2020) | 23.3 | 8.6 | 9.7 | 13.9 |
| TENT (Wang et al., 2021) | 26.5 | 8.7 | 10.5 | 15.2 |
| EATA (Niu et al., 2022a) | 27.9 | 8.7 | 10.5 | 15.7 |
| MonoTTA (Lin et al., 2024) | 32.0 | 9.2 | 9.7 | 17.0 |
| SPA-I | 32.6 | 10.1 | 8.8 | 17.2 |
| SPA | 34.4 | 11.1 | 9.7 | 18.4 |

*Table 12.* Effects of combined *vs.* separate augmentation strategy in SPA on 3D monocular detection with MonoFlex as source model. **SPA-I** applies low-frequency amplitude mask (Eqn. (4)) and high-frequency noise injection (Eqn. (5)) in a single image simultaneously, obtaining one augmented image. While SPA augments an image using Eqn. (4) and Eqn. (5) separately, generating two augmented images for test-time self-bootstrapping learning.

| | KITTI-Fog ($AP_{3D|R40}$, %, ↑) | | | |
|---|---|---|---|---|
| Method | Car | Pedestrian | Cyclist | Avg. |
| No Adapt | 7.8 | 2.0 | 3.6 | 4.5 |
| BN Adapt (Schneider et al., 2020) | 23.3 | 8.6 | 9.7 | 13.9 |
| TENT (Wang et al., 2021) | 26.5 | 8.7 | 10.5 | 15.2 |
| EATA (Niu et al., 2022a) | 27.9 | 8.7 | 10.5 | 15.7 |
| MonoTTA (Lin et al., 2024) | 32.0 | 9.2 | 9.7 | 17.0 |
| *Self-bootstrapping learning of our SPA by aligning:* | | | | |
| Reg. heads | 32.4 | 8.7 | 8.7 | 16.6 |
| Cls. heads | 32.2 | 10.7 | 9.3 | 17.4 |
| Reg. heads & Cls. heads | 34.4 | 11.1 | 9.7 | 18.4 |

*Table 13.* Effects of SPA aligning different heads' predictions (classification and regression heads) in 3D monocular object detection. We use MonoFlex as the source model.

| Method | Average Accuracy (%) |
|---|---|
| Source | 31.4 |
| CoTTA (Wang et al., 2022) | 34.0 |
| EATA (Niu et al., 2022a) | 44.4 |
| DeYO (Lee et al., 2024) | 45.9 |
| ROID (Marsden et al., 2024) | 46.8 |
| CMF (Lee & Chang, 2024) | 48.1 |
| SPA (ours) | 49.2 |
| SPA + Tent (ours) | 50.9 |
| SPA + EATA (ours) | 52.5 |

*Table 14.* Comparisons on ImageNet-C (level 5) with ResNet-50.

constructed weak-to-strong learning signals at different frequency ranges independently. In this section, we further compare SPA with SPA-I, which simultaneously applies two augmentation strategies in a single image. From results in Tables 11 and 12, SPA-I performs slightly worse than SPA but still achieves better or comparable performance compared to prior SOTAs, suggesting its superiority.

**Adaptation Efficiency** Notably, though SPA involves one more forward and backward propagation, it remains efficient and operates in real-time, achieving 79 FPS (*vs.* SPA-I: 125 FPS) on a single A100 GPU with ViT-Base and ImageNet-C. Here, SPA-I is the variant that applies two deteriorations within a single image (1 augmentation) and achieves similar ImageNet-C accuracy: SPA (70.1%) vs SPA-I (69.0%) as in Table 11. SPA is more efficient than prior augmentation-based methods such as MEMO (Zhang et al., 2022), CoTTA (Wang et al., 2022), and TPT (Shu et al., 2022), which require 64, 34(or 2), and 63 augmentations per sample, respectively. It also matches the efficiency of entropy-based SAR. The FPS on ImageNet-C with ViT-Base (on a single A100 GPU) are: SPA-I (125) > SAR (102) > SPA (79) > CoTTA (36).

**Effectiveness of Aligning Regression Heads in SPA for 3D Monocular Object Detection** The 3D monocular detection task comprises both classification heads to identify *object class* within each 3D bounding box, and regression heads to predict the 3D bounding box *coordinates*, *dimensions*, *depths*, and *angles* which provides a comprehensive spatial understanding of each detected object. However, existing TTA methods (Schneider et al., 2020; Wang et al., 2021; Niu et al., 2022a; Lin et al., 2024) for 3D MonoDet focus on designing TTA loss on classification heads while overlooking the regression heads with rich predictions. In contrast, SPA is task-agnostic, making it applicable to both classification and regression heads seamlessly. In SPA, we demonstrate that leveraging the regression heads (with rich spatial information) for TTA, *i.e.*, aligning prediction consis-

tency of regression predictions, offers rich learning signals at test time. As shown in Table 13, SPA achieves comparable performance to existing methods using only regression self-supervision, *e.g.*, with an average $AP_{3D|R40}$ of 16.6% (SPA with Reg. Heads Alignment) *vs.* 15.7% (EATA). When further incorporating classification supervision, our SPA is able to surpass the existing state-of-the-art method, MonoTTA, which focuses only on the classification head, by an average of 1.4%. These results collectively highlight the effectiveness of SPA, and underscore the importance of exploiting both information from regression and classification predictions to design more general and effective TTA solution.

**Effectiveness of SPA with Different Model Architectures** In our experiments, we use the ViT-Base model for image classification, MonoFlex (Zhang et al., 2021) for 3D monocular object detection, and Segformer-B5 (Xie et al., 2021) for segmentation tasks. Here, ViT-Base and Segformer-B5 employ transformer-based architectures, while MonoFlex is based on convolutional neural networks. Our results in Tables 1, 2, 3 and 4 demonstrate that SPA performs well on all three backbone models, demonstrating its generality across different types of model architectures.

**More Results on Classification with ResNet-50** In this section, we provide more experiments on ResNet-50 with ImageNet-C. As shown in Table 14, the improvements observed on ResNet-50 are consistent with our experiments on ViT-Base. This further underscores SPA's effectiveness both as a standalone method and as a plug-and-play module to enhance existing methods.

