# OpenReview forum: "Self-Bootstrapping for Versatile Test-Time Adaptation"
_ICML.cc/2025/Conference — ICML 2025 poster_

### Official Review · Reviewer_tvoR · 2025-03-11

**Overall Recommendation:** 3

**Summary:**

This paper proposes Self-bootstrapping for versatile test-time adaptation, a general TTA framework that adapts models across classification, regression, and dense prediction tasks without requiring source data. SPA introduces weak-to-strong self-bootstrapping learning, ensuring adaptation by aligning a deteriorated (weak) view’s predictions with the original (strong) view using Fourier-based augmentations—low-frequency amplitude masking and high-frequency noise injection.

**Claims And Evidence:**

Yes

**Essential References Not Discussed:**

A lot of TTA method for image classification missed:

1. Roid, published in WACV 2024;
2. ViDA, published in ICLR 2024;
3. A Versatile Framework for Continual Test-Time Domain Adaptation: Balancing Discriminability and Generalizability, published in CVPR 2024;
4. CMF, published in ICLR 2024;
5. SLWI, ICML 2024, etc.

For object detection:
1. IoU-filter published in CVPRW 2024
2. Memclr published in wacv 2023

**Experimental Designs Or Analyses:**

Many new TTA works on image classification were published in 2024, and comparing SPA with them on CIFAR-10-C, CIFAR-100-C, DomainNet, and additional architectures (e.g., WideResNet, ResNet) would strengthen its evaluation.

For segmentation, the use of "rounds" is unclear—if the same test samples appear multiple times, this is not true online TTA; if not, the sample split criteria per corruption type need clarification.

For object detection, the MemCLR (WACV 2023) method might be considered to provide a more complete comparison.

**Methods And Evaluation Criteria:**

Almost make sense.

This paper aims to demonstrate that the proposed Fourier-based augmentations work across various image-based tasks, including classification, segmentation, and 3D detection. However, the evaluation could benefit from a clear primary focus, such as image classification, to better align with existing TTA benchmarks. Expanding the evaluation to include CIFAR-10-C and CIFAR-100-C datasets and incorporating additional backbones like WideResNet and ResNet would strengthen the comparison with prior TTA methods and provide a more comprehensive assessment of generalizability.

**Other Comments Or Suggestions:**

See the comments.

**Other Strengths And Weaknesses:**

Strengths:

- The Fourier-based augmentation strategy has potential for broader applications in various image-based tasks beyond TTA.

- The paper evaluates the method across different image-based tasks (classification, segmentation, and detection), demonstrating its generalizability.

Weaknesses:

- The method's complexity is limited, as it primarily modifies the augmentation strategy within an existing consistency-based TTA framework, which may not be a major conceptual leap.

- The experiments lack a clear focus, spreading across multiple tasks but not providing in-depth validation on any single one.

- The high-/low-frequency augmentation approach could benefit from stronger empirical analysis or theoretical justification to validate its effectiveness.

- Adaptation efficiency and latency are not discussed, which are crucial for real-world TTA deployment.

- Additional comparisons with recent TTA methods (e.g., 2024 TTA classification works, MemCLR for detection) would strengthen the evaluation.

**Questions For Authors:**

Have you explored the effectiveness of this augmentation strategy on Vision-Language Models (VLMs) like CLIP and SigLIP?

Your experiments include continual adaptation for segmentation—does the strategy also work for continual or mixed-domain TTA in image classification?

How does the computational cost of the proposed Fourier-based augmentations compare to standard TTA augmentations? Will it slow the adaptation time?

**Relation To Broader Scientific Literature:**

This paper extends TTA by adapting test samples into a weak-to-strong self-bootstrapping framework for stable adaptation. SPA introduces Fourier-based augmentations—low-frequency masking and high-frequency noise injection—to provide stronger adaptation signals while preserving geometric structure. This augmentation is general across classification, segmentation, and detection tasks, and might benefit to supervised learning (not verified)

**Theoretical Claims:**

No theoretical claim and proofs.

---

> ### Author Rebuttal · Authors · 2025-04-01
>
> >Q1. Results on CIFAR-100 and ResNet
>
> In the submission, we compare SPA on ImageNet-C/R/Sketch/A as they are large-scale, more challenging than CIFAR-100, and commonly used. We now report more results on CIFAR100-C below to further validate SPA. **Pls see Reviewer 61cM's Q1 for more ResNet results**.
> |Method|Avg Acc on CIFAR100-C|
> |-|:-:|
> |CoTTA|69.6|
> |EATA|72.1|
> |ActMAD|74.0|
> |DeYO|74.0|
> |ROID|76.3|
> |SPA(Ours)|75.4|
> |SPA+ActMAD+Tent|77.0|
> |SPA+ROID|77.4|
> >Q2. Clarification on "rounds" in segmentation exps
>
> “Round” refers to the number of repeated sequence of datasets to simulate long-term continual TTA (e.g., Fog-Night-Rain-Snow-Fog-…). This is a standard evaluation protocol in continual TTA following CoTTA. We will make this clearer.
> >Q3. Comparison with more baselines
>
> As you suggested, we compare SPA with more baselines in the table below on classification, showing our efficacy. For detection, we acknowledge that the suggested baselines are designed for 2D tasks, while we focus on more challenging 3D monocular object detection. Due to time limits, it is non-trivial to implement them in our 3D setting. Here, we would like to clarify that MonoTTA (ECCV'24) is tailored for 3D detection and is the current SOTA (more latest than suggested ones), making it more relevant for comparison. Thus, we believe the current thorough comparisons with MonoTTA have fully validated SPA's effectiveness. We will discuss all mentioned methods in the revision.
>
> Table. More results on ImageNet-C with ViT-B
> |Method|Avg Acc|
> |-|-|
> |ROID|68.0|
> |ViDA|60.8|
> |CMF|69.2|
> |SPA(Ours)|70.1|
> |SPA+ActMAD|71.2|
> |SPA+ActMAD+TENT|71.6|
> >Q4. The method's complexity is limited, ...
>
> SPA's main contribution lies in its overall design for versatile online fully TTA. It comprises an active, deterioration-driven self-bootstrapping scheme and geometry-preserving augmentations inspired by our Fourier analysis of domain shifts.
>
> Unlike prior methods enforcing consistency across different augmentations with a teacher-student scheme, our SPA generates a deteriorated image to create an information gap, enabling self-distillation *in a single model* from strong to weak predictions. The distilled knowledge is then directly fed back to the strong branch via shared parameters, forming a closed-loop TTA process.
>
> Though simple in form, SPA’s insights and overall design are non-trivial and challenging, reflected in the result of extensive preliminaries where direct consistency learning or traditional augmentations failed to deliver satisfactory performance. We also believe such method simplicity helps enhance both usability and impact.
> >Q5. The experiments lack a clear focus, ...
>
> Our goal is to design a versatile TTA method, so we evaluate SPA across multiple tasks. Actually, our validations are thorough: we verified our superiority as standalone or as a plug-and-play module on different tasks, e.g., outperforming prior SOTAs like DeYO (ICLR'24) in classification and MonoTTA (ECCV'24) in 3D detection. We also provided extensive ablations and analyses for different tasks. As you suggested, we also compare with more baselines to verify our efficacy, pls see Q3 and Q9.
> >Q6. The method could benefit from stronger empirical analysis to validate its effectiveness
>
> In Figure 2, we empirically analyze how domain shifts manifest in Fourier domain, and derive insights of which image deteriorations can supply sufficient learning signals for our self-bootstrapping. Extensive experiments (Tables 5,6,7) further validate the conclusions drawn from Figure 2. These analyses are also positively acknowledged by other reviewers, e.g., *“analyses…particularly insightful”* [DdZn] and *“sufficiently supporting the conclusions drawn”* [QKc5].
> >Q7. Adaptation efficiency
>
> Our SPA is much more efficient than augmentation-based CoTTA, which uses 2 or 34 augmentations per sample, while SPA needs only 1–2. It also matches the efficiency of entropy-based SAR. FPS on ImageNet-C with ViT-Base (on A100) are: SPA-I(125)>SAR(102)>SPA(79)>CoTTA(36). Here, SPA-I is a variant that applies two deteriorations within a single image (1 aug) and achieves similar ImageNet-C acc: SPA(70.1%) vs SPA-I(69.0%). Partial results were in Appendix C and we will make this clearer.
>
> >Q8. Efficacy on VLM
>
> Our SPA works on VLM, pls see Reviewer DdZn's Q4.
> >Q9. Results under continual or mixed-domain TTA in classification
>
> Our SPA works under such scenarios, pls see Reviewer DdZn'Q3.
> >Q10. Computation cost of augmentation
>
> Our augmentations are efficient: 1) Fourier masking uses PyTorch's FFT/iFFT, accelerated via parallelized GPU operations; 2) Noise injection is a trivial element-wise addition. Augmentation takes only a tiny (negligible) fraction of the total adaptation time, and the FPS of SPA is almost the same whether using our augs or MoCo/SimCLR’s.
>
> We deeply appreciate your constructive comments. We sincerely hope our clarifications above have addressed your questions and can improve your opinion of our work.

---

> > ### Comment · Reviewer_tvoR · 2025-04-04
> >
> > I saw the results on CLIP, the Imagenet-r is for the ood dataset benchmarks. I am wondering:
> >
> > 1. if this could work on cross-dataset benchmark.
> > 2. the setups of the CLIP experiments

---

> > > ### Author Response · Authors · 2025-04-04
> > >
> > > Dear Reviewer tvoR,
> > >
> > > Thanks for your reply. This is a valuable question, and we would like to answer it below.
> > >
> > > **$\bullet$ The setups of CLIP experiments:**
> > >
> > > We freeze the text encoder of the CLIP model and treat it as a fixed classifier. Our method is then applied to the image encoder, where we adapt the affine parameters of its normalization layers.
> > >
> > > **$\bullet$ On cross-dataset benchmarks:**
> > >
> > > Our method primarily focuses on addressing the **OOD issue in the visual modality** through our deterioration-driven self-bootstrapping learning, by **improving the visual feature representation**.
> > >
> > > However, for commonly used **cross-dataset benchmarks on CLIP model**, the image distributions are often relatively more stable compared to OOD datasets, and the image encoder already provides more semantical representations. The **performance bottleneck instead often lies in the text modality (i.e., the text encoder)**—specifically, the quality of the classifier formed from text embeddings. This is also supported by 1) prior VLM TTA methods, such as TPT and C-TPT, which focus on adapting the text branch to improve cross-dataset generalization, and 2) prior visual modality-focused TTA methods (like Tent and ETA) achieve limited performance gains in this cross-dataset scenario.
> > >
> > > Therefore, the performance gain of our SPA method on cross-dataset scenarios with CLIP is not as competitive as its improvement on OOD scenarios—but it still provides benefits—as we do not adapt the text branch.
> > >
> > > Extending our method to also adapt the text branch to further boost performance in cross-dataset scenarios is an interesting and promising direction. We leave this for future work, and believe that our current focus on addressing visual OOD issues already makes meaningful contributions, as this is a fundamental challenge for numerous vision-involved models.
> > >
> > > Table. Results on CLIP-ViT-B for cross-dataset generalization.
> > > |Method \ Dataset|DTD|UCF101|Aircraft|Avg. Acc. (%)|
> > > |-|:-:|:-:|:-:|:-:|
> > > |Source|44.3|65.1|23.8|44.4|
> > > |VLM TTA methods:
> > > |TPT|46.7|67.3|23.4|45.8|
> > > |C-TPT|46.0|65.7|24.0|45.2|
> > > |Vision TTA methods:
> > > |Tent|45.2|66.0|23.4|44.9|
> > > |ETA|44.7|66.1|23.7|44.8|
> > > |SAR|44.6|66.5|23.4|44.8|
> > > |DeYO|44.2|66.0|22.7|44.3|
> > > |SPA (ours)|45.4|66.2|23.6|45.1|
> > >
> > > We greatly appreciate your invaluable reviews to improve the quality of our paper and your insightful further comments on cross-dataset evaluation. We sincerely hope our clarifications above have addressed your questions and can improve your opinion of our work. We are happy to continue the discussion if you have further questions.
> > >
> > > Best,
> > >
> > > The Authors
> > >
> > > ### Post Discussion Update：
> > >
> > > Dear Reviewer tvoR,
> > >
> > > Thank you for upgrading your score! Your invaluable review comments are immensely beneficial in improving the quality of our paper!
> > >
> > > Best,
> > >
> > > The Authors

---

### Official Review · Reviewer_QKc5 · 2025-03-13

**Overall Recommendation:** 4

**Summary:**

The paper explores the impact of typical distribution shifts on the information content of images across different spatial frequencies in the Fourier domain. It highlights that low-frequency components dominate in terms of information power, and removing these components provides more effective learning signals compared to masking high-frequency components. Based on this observation, the authors propose a data augmentation strategy that involves randomly masking the low-frequency amplitudes of an image in the Fourier domain. Additionally, they inject noise into the image to enhance the information power of high-frequency components, compensating for the loss of learning signals at lower frequencies. Experimental results demonstrate that this method, whether applied on its own or as a plug-and-play module, improves performance in tasks such as classification, segmentation, and 3D monocular detection across both transformer-based and CNN-based models.

## update after rebuttal
Thanks to the authors for their response. I do not have any further queries.

**Claims And Evidence:**

The main claim of this paper centers around the validity of the proposed data generation scheme, supported by experiments. Table 1 provides evidence for classification tasks, Table 3 validates the approach on object detection, and Table 4 focuses on segmentation. Additionally, Table 5 includes an ablation study that appears to substantiate the points made, sufficiently supporting the conclusions drawn. However, the novelty of the SPA approach may be somewhat overstated in the paper; it presents ideas that are not particularly new.

**Essential References Not Discussed:**

N/A

**Experimental Designs Or Analyses:**

The experiments are comprehensive, with no gaps in the ablation studies. All argued applications have been validated through experimentation.

**Methods And Evaluation Criteria:**

The TTA benchmark is well-established, and this paper runs most of the convincing datasets, with a sufficiently large testing scale to support its claims.

**Other Comments Or Suggestions:**

Please emphasize augmentation more than SPA in future versions.

**Other Strengths And Weaknesses:**

This paper is well-written.

**Questions For Authors:**

N/A

**Relation To Broader Scientific Literature:**

N/A

**Theoretical Claims:**

The augmentation scheme proposed in this paper relies more on empirical experimental validation. While Figure 2 provides some insightful theoretical basis, it is not sufficiently theoretical to fully support the proposed method.

---

> ### Author Rebuttal · Authors · 2025-04-01
>
> We deeply appreciate your valuable feedback and your recognition of the novelty and contributions of our work for designing a challenging versatile fully TTA framework. Our SPA incorporates several components, including an active, deterioration-driven self-bootstrapping scheme (distinct from feature-level BYOL), as well as carefully crafted insights into geometry-preserving augmentation strategy design. The overall design of SPA with these innovations is challenging and non-trivial based on our analysis. These innovations collectively ensure the stability and effectiveness of SPA for online fully test-time adaptation. Thank you again for your insightful suggestion. We will make the novelty and contributions of our work clearer in the revised version.

---

### Official Review · Reviewer_DdZn · 2025-03-14

**Overall Recommendation:** 4

**Summary:**

The paper introduces Self-Bootstrapping for versatile Test-Time Adaptation (SPA), a novel framework that enables TTA across multiple tasks—classification, segmentation, and 3D detection. The authors propose a geometry-preserving augmentation strategy using low-frequency amplitude masking and high-frequency noise injection in the Fourier domain. This approach maintains spatial structure while providing sufficient learning signals for adapting models at test time without source data access.

**Claims And Evidence:**

- The core claim of versatility across task types is well-supported through experiments on ImageNet-C/R/A/Sketch, KITTI-C, and ACDC datasets.
- Results demonstrate consistent improvements over baseline methods across different architectures (ViT, CNN) and tasks.
- The ablation studies effectively validate design choices, particularly regarding frequency-domain augmentation strategies.

**Essential References Not Discussed:**

Diffusion-TTA - https://diffusion-tta.github.io/
TTT-MAE - https://papers.neurips.cc/paper_files/paper/2022/file/bcdec1c2d60f94a93b6e36f937aa0530-Paper-Conference.pdf

**Experimental Designs Or Analyses:**

- Experimental setup is sound with appropriate baselines.
- Missing comparison with certain TTA methods: Notable omissions include MT3, TTT_MAE, and Diffusion-TTA, which would strengthen the evaluation.
- The analyses of how different domain shifts manifest in the frequency domain are particularly insightful.




Diffusion-TTA - https://diffusion-tta.github.io/
TTT-MAE - https://papers.neurips.cc/paper_files/paper/2022/file/bcdec1c2d60f94a93b6e36f937aa0530-Paper-Conference.pdf

**Methods And Evaluation Criteria:**

- The proposed frequency-domain analysis and augmentation approach is novel and well-motivated.
- Benchmarks cover a good range of distribution shifts and tasks.
- Evaluation methodology follows established protocols in the field.

**Other Comments Or Suggestions:**

Nothing in particular

**Other Strengths And Weaknesses:**

Strengths:

- The approach's versatility across regression and classification tasks is impressive
- Geometry-preserving augmentations address a key limitation in prior work
- Functions well as both standalone method and plug-and-play module

Weaknesses:

- No comparison with TTT-series methods that might be competitive, i would prefer if the authors could do apples to apples comparision against MT3, so that we can use directly the numbers reported in their paper
- I'm still missing a good intution/analysis as to why conventional augmentations are not good, although Figure 2 shows RAPSD changes with different shifts, i'm still struggling to understand why standard augmentations such as Random Masking or Blur are not sufficient?

**Questions For Authors:**

- Why were TTT methods like MT3, TTT_MAE and Diffusion-TTA not included in the comparisons? These seem like competitive approaches worth benchmarking against.
- How does SPA perform on more realistic distribution shifts beyond those tested? like for instance objectnet
- Have you explored potential integration with foundation models?


[1] Objectnet - https://papers.nips.cc/paper_files/paper/2019/hash/97af07a14cacba681feacf3012730892-Abstract.html

**Relation To Broader Scientific Literature:**

The paper builds upon and extends several research directions in domain adaptation and self-supervised learning. It draws conceptually from self-supervised contrastive methods like BYOL and DINO but adapts these approaches specifically for test-time adaptation. The authors' frequency-domain augmentation strategy connects to a growing body of work on Fourier-domain analysis in computer vision, though they uniquely apply it to preserve geometric structure for dense prediction tasks. The paper sits at the intersection of consistency-based TTA methods (like MEMO), entropy-based approaches (TENT, EATA), and structure-preserving adaptation—extending beyond prior work by creating a unifying framework applicable to a wider range of tasks.

**Theoretical Claims:**

No formal proofs are presented.

---

> ### Author Rebuttal · Authors · 2025-03-31
>
> We deeply appreciate your valuable feedback and constructive comments on improving the quality of our paper. We would like to address your questions below.
>
> >Q1. Comparison with TTT-series methods.
>
> Thank you for your suggestion. We follow the comparison setting used in Diffusion-TTA and directly compare our results with the TTT methods reported in their work, by implementing our method using their adopted model and experimental environment. This is because TTT-series methods modify the model training process, making them relatively difficult to implement under our diverse environmental setups. As shown in Table A, our SPA, as a fully TTA method, still achieves superior performance, further demonstrating its effectiveness. We will discuss all mentioned TTT-series methods in the revision.
>
> Table A. Comparisons with TTT-series methods on ImageNet-C w.r.t. acc (%).
> |Model+Method|gauss|fog|pixel|snow|contrast|Avg|
> |-|:-:|:-:|:-:|:-:|:-:|:-:|
> |Customized ViT-L/16 classifier|17.1|38.7|47.1|35.6|6.9|29.1|
> | + TTT-MAE|37.9|51.1|65.7|56.5|10.0|44.2|
> |ViT-B/32|39.5|35.9|55.0|30.0|31.5|38.4|
> | + Diffusion-TTA|46.5|56.2|64.7|50.4|33.6|50.3|
> | + SPA (Ours)|49.4|56.9|66.7|49.9|51.5|54.9|
>
> >Q2. Why are conventional augmentations not good?
>
> Conventional augmentations can be broadly categorized into two groups based on whether they preserve geometric information.
>
> - **Geometry-preserving augmentations** (e.g., grayscale conversion, brightness, contrast, blur) typically adjust the global distribution of images—for instance, changing brightness by adding the same constant across all pixels. Such augmentation patterns are relatively simple and lack diversity or uniqueness for different samples, thus providing limited learning signals for our self-bootstrapping learning pipeline. As in Table 6, none of these augmentations, individually or combined, effectively provide TTA with rich learning signals. These augmentations are also often sensitive to the corruption type and struggle to perform stably across all corruptions, leading to limited overall performance.
> - **Non-geometry-preserving augmentations** (e.g., random resizing, cropping, image masking, SimCLR, MoCo, AugMix) introduce randomness for different samples, which randomly deteriorate the local information of different images, thus can provide more diverse learning signals for SPA. However, these augmentations shall disrupt the image’s geometric structure and directly desert the potentially useful image information. As in Table 7, SPA with these augmentations achieves considerable performance on image classification but performs poorly on finer dense prediction tasks like 3D monocular detection. In contrast, our Fourier augmentations preserve the overall information and geometry to support dense prediction tasks and also introduce sufficient diversity for different samples, supplying rich signals for SPA and achieving improved adaptation performance.
>
> >Q3. Results on more realistic distribution shift scenarios.
>
> Thank you for your valuable suggestion. We have conducted additional experiments on ObjectNet in Table B, based on the codebase of VisDA-2021 challenge. Furthermore, we also evaluate SPA under more 'realistic settings', including continual adaptation (Table C) and mixed domain adaptation (Table D). The results across all these tables further validate the effectiveness of SPA, both as a standalone method and as a plug-and-play module to enhance existing approaches.
>
> Table B. Results on ObjectNet with ResNet-50.
> |Method|Source|Tent|ETA|SAR|DeYO|ROID|SPA(ours)|SPA+ETA(ours)|
> |-|-|-|-|-|-|-|-|-|
> |Acc (%)|27.2|27.5|26.0|28.6|29.8|28.7|30.9|32.3|
>
> Table C. Results of continual adaptation on ImageNet-C (level 5) with ViT-Base. We report avg acc of 15 corruptions.
> |Method|Avg Acc (%)|
> |-|-|
> |Source|55.5|
> |EATA|67.3|
> |SAR|61.3|
> |ROID|67.9|
> |ActMAD|59.9|
> |SPA (ours)|68.6|
> |SPA+ActMAD (ours)|71.5|
> |SPA+ActMAD+Tent (ours)|71.8|
>
> Table D. Results on mixture of 15 corruptions of ImageNet-C (level 5) with ViT-Base.
> |Method|Acc (%)|
> |-|-|
> |Source|55.5|
> |EATA|63.7|
> |SAR|60.7|
> |DeYO|55.1|
> |ROID|62.0|
> |SPA (ours)|67.1|
> |SPA+ActMAD (ours)|68.2|
> |SPA+ActMAD+Tent (ours)|68.5|
>
> >Q4. Integration with foundation models.
>
> We briefly test our method on CLIP-ViT model and report results in Table E. With CLIP, our method still works, further suggesting the versatility of SPA.
>
> Table E. Results with CLIP-ViT-B on ImageNet-R.
> |Method|Source|TPT|C-TPT|ETA|SAR|DeYO|SPA (ours)|SPA+ETA (ours)|
> |-|-|-|-|-|-|-|-|-|
> |Acc (%)|74.0|77.1|76.0|76.9|75.6|76.6|77.2|78.2|
>
> We thank you for appreciating our contributions. We sincerely hope our clarifications above have addressed your questions and can improve your opinion of our work.

---

> > ### Comment · Reviewer_DdZn · 2025-04-07
> >
> > Thanks for the new experiments, I hope the authors include the new experiments and the new baselines in the final version of the paper. Also it will be good to make a clear distinction between TTT methods that require specialized model training like TTT-MAE and methods such as Diffusion-TTA that do not. As it's unclear to me what falls under the TTT umbrella.
> >
> > I have increased my rating.

---

> > > ### Author Response · Authors · 2025-04-08
> > >
> > > Dear Reviewer DdZn,
> > >
> > > We are super glad and appreciate that you have increased your score! We will include all these new results and discussions in our revised paper.
> > >
> > > The differences between TTT method (TTT-MAE), Diffusion-TTA, and our proposed SPA are: 1) TTT-MAE modifies the original model training process; 2) Diffusion-TTA does not alter the original training process, but it relies on an additional pre-trained diffusion model and jointly trains both the discriminative model (which requires adaptation) and the diffusion model during the testing phase; 3) our SPA does not need additional model and can be directly applied to a single (discriminative) model for adaptation. We will make these distinctions clearer in our revision.
> > >
> > > ||Modify original training process?|Rely on additional pre-trained (diffusion) model? |Model learned at test time|
> > > |-|-|-|-|
> > > |TTT-MAE|Yes|No|Discriminative model|
> > > |Diffusion-TTA|No|Yes|Discriminative model and Diffusion model|
> > > |SPA (Ours)|No|No|Discriminative model|
> > >
> > > Thank you again for your invaluable review comments on improving the quality of our work!
> > >
> > > Best,
> > >
> > > The Authors

---

### Official Review · Reviewer_61cM · 2025-03-14

**Overall Recommendation:** 3

**Summary:**

This paper proposes an image augmentation strategy utilizing the Fourier domain for randomly masking the low-frequency amplitude of an image. Further, it augments the image with noise injection to account for the lack of learning signals at high frequencies.
The paper reports experimental results on classification, segmentation, and 3D monocular detection tasks to show the effectiveness of the proposed approach.

**Claims And Evidence:**

The claims made in the submission are supported by the improved performance in the experiments.

**Essential References Not Discussed:**

A Fourier transform for domain adaptation that also involves masking has already been proposed by [1].
This limits the novelty of the proposed approach utilizing Fourier transform with masking for TTA, which is a setting related to domain adaptation.

**References**
1. Yang, Yanchao, and Stefano Soatto. "Fda: Fourier domain adaptation for semantic segmentation." Proceedings of the IEEE/CVF conference on computer vision and pattern recognition. 2020.

**Experimental Designs Or Analyses:**

Refer to Questions and Weaknesses.

**Methods And Evaluation Criteria:**

The benchmark datasets make sense but it would better a more fair comparison if experiments on ImageNetC is on ViT, and not ResNet-50 like prior works such as CoTTA [1], PETAL [2], EcoTTA [3], etc.

**References**
1. Wang, Qin, et al. "Continual test-time domain adaptation." Proceedings of the IEEE/CVF Conference on Computer Vision and Pattern Recognition. 2022.
2. Brahma, Dhanajit, and Piyush Rai. "A probabilistic framework for lifelong test-time adaptation." Proceedings of the IEEE/CVF Conference on Computer Vision and Pattern Recognition. 2023.
3. Song, Junha, et al. "Ecotta: Memory-efficient continual test-time adaptation via self-distilled regularization." Proceedings of the IEEE/CVF Conference on Computer Vision and Pattern Recognition. 2023.
4. Yang, Yanchao, and Stefano Soatto. "Fda: Fourier domain adaptation for semantic segmentation." Proceedings of the IEEE/CVF conference on computer vision and pattern recognition. 2020.

**Other Comments Or Suggestions:**

Please include the reference and highlight the novelty or difference between [1] and the proposed approach in the paper (appendix if the space is limited).

**References**
1. Yang, Yanchao, and Stefano Soatto. "Fda: Fourier domain adaptation for semantic segmentation." Proceedings of the IEEE/CVF conference on computer vision and pattern recognition. 2020.

**Other Strengths And Weaknesses:**

**Strengths**
* Performing image augmentation by utilizing Fourier transform for TTA is an interesting idea
* Experimental gains demonstrate the effectiveness of the proposed approach

**Weaknesses**
* Fourier transform for domain adaptation that also involves masking has already been proposed by [4].
* Experiments on ImageNetC is on ViT, and not ResNet-50 like prior works such as CoTTA [1], PETAL [2], EcoTTA [3], etc.

**References**
1. Wang, Qin, et al. "Continual test-time domain adaptation." Proceedings of the IEEE/CVF Conference on Computer Vision and Pattern Recognition. 2022.
2. Brahma, Dhanajit, and Piyush Rai. "A probabilistic framework for lifelong test-time adaptation." Proceedings of the IEEE/CVF Conference on Computer Vision and Pattern Recognition. 2023.
3. Song, Junha, et al. "Ecotta: Memory-efficient continual test-time adaptation via self-distilled regularization." Proceedings of the IEEE/CVF Conference on Computer Vision and Pattern Recognition. 2023.
4. Yang, Yanchao, and Stefano Soatto. "Fda: Fourier domain adaptation for semantic segmentation." Proceedings of the IEEE/CVF conference on computer vision and pattern recognition. 2020.

**Questions For Authors:**

1. How is the noise factor γ hyperparameter tuned? Is the performance on the same test data monitored to get the best hyperparameters?
2. A Fourier transform for domain adaptation that also involves masking has already been proposed by [1]. This limits the novelty of the proposed approach utilizing Fourier transform with masking for TTA, which is a setting related to domain adaptation. Can the authors elaborate on how the proposed approach is a non-trivial extension of this prior work?

**References**
1. Yang, Yanchao, and Stefano Soatto. "Fda: Fourier domain adaptation for semantic segmentation." Proceedings of the IEEE/CVF conference on computer vision and pattern recognition. 2020.

**Relation To Broader Scientific Literature:**

The key contributions of this paper can lead to better test-time adaptation in general. However, the novelty seems limited, as pointed out in "Essential References Not Discussed" and "Questions".

**Theoretical Claims:**

This work does not involve theoretical claims or proofs.

---

> ### Author Rebuttal · Authors · 2025-04-01
>
> We deeply appreciate your valuable feedback and constructive comments on improving the quality of our paper. We would like to address your questions below.
>
> >Q1. More classification results on ResNet-50.
>
> Thanks for your suggestion. We conduct additional experiments on ResNet-50 and compare with more SOTA approaches. As in the table below, the improvements observed on ResNet-50 are consistent with our experiments on ViT-Base. This further underscores SPA's effectiveness both as a standalone method and as a plug-and-play module to enhance existing methods.
>
> |Method|Avg Acc (%) over 15 corruptions of ImageNet-C|
> |-|:-:|
> |NoAdapt (ResNet-50)|31.4|
> |CoTTA|34.0|
> |EATA|44.4|
> |DeYO|45.9|
> |ROID|46.8|
> |CMF|48.1|
> |SPA (ours)|49.2|
> |SPA+Tent (ours)|50.9|
> |SPA+EATA (ours)|52.5|
>
> >Q2. Differences from FDA [A].
>
> Thank you for pointing out this related work. While FDA also utilizes the Fourier augmentation, SPA's primary contributions lie in the overall design of a versatile TTA scheme. SPA comprises several key components, including an active, deterioration-driven self-bootstrapping scheme (distinct from BYOL), and geometry-preserving augmentations inspired by our Fourier analysis across domain shifts. All these innovations are non-trivial and collectively contribute to SPA’s effectiveness and versatility. Compared with FDA, which focuses on unsupervised domain adaptation, our SPA method mainly differs in the following aspects.
>
> - **Different Augmentations:** FDA augments a source domain image by replacing the low-frequency spectrum of the source image with that from a target image. This augmentation relies on paired source and target data, which is infeasible in TTA as only a single unlabeled target sample is available at test time. Unlike FDA, SPA randomly masks the low-frequency spectrum and injects noise for a single target image, without requiring the source images.
> - **Different Methods:** FDA uses Fourier augmentation to transfer the source image style to the target domain, creating a target-style labeled source dataset $D^{s\rightarrow t}$. It then jointly trains the model offline using both $D^{s\rightarrow t}$ and the original target dataset $D^{t}$ to mitigate domain shifts. In contrast, SPA targets the more general and challenging setting of fully online TTA. It establishes a versatile self-bootstrapping framework that performs active weak-to-strong learning from a deteriorated view to the original image, for each given target test sample.
> - **Different Design Motivations:** The key motivation of FDA is style transfer（$s\rightarrow t$）for cross-domain training, by swapping the spectrum in Fourier domain. Unlike FDA, the online unsupervised learning in SPA is highly unstable, and thus our key motivation is how to design effective deterioration/augmentation strategies to conquer this. To this end, we analyze how information power shifts across frequencies under domain shift and leverage these insights to design augmentations that consistently deliver informative signals across all frequencies for SPA, meanwhile maintaining stability.
>
> We will include the above discussions in the revised paper.
>
> [A] Fda: Fourier domain adaptation for semantic segmentation. CVPR 2020.
>
> >Q3. How is the noise factor $\gamma$ hyperparameter tuned?
>
> We did not carefully tune $\gamma$ on each individual test set as at test time we do not have ground truth labels. In main experiments, we briefly set $\gamma$ to 0.4 for all classification datasets, including ImageNet-C (15 corruptions, ImageNet-R/Sketch/A), and $\gamma=0.1$ for all fine-grained tasks of detection and segmentation, including KITTI-C (13 corruptions) and Cityscape-to-ACDC.
>
> We also show the sensitivity of $\gamma$ in Figure 3 on a single corruption (test set) of ImageNet-C(Gaussian) and KITTI-C(Fog). From the results, $\gamma\in[0.1,0.5]$ works stably on classification, while the stable range of $\gamma$ on mono 3D detection is relatively narrower, with performance slightly declining when $\gamma>0.2$—though it still remains significantly better than without adaptation. This difference arises because, for classification, the task is at the image level and does not strictly require content invariance, allowing for a higher $\gamma$ to provide richer learning signals. In contrast, 3D detection involves dense predictions where high noise levels could significantly disrupt the original image content, making it challenging for our self-bootstrapping learning.
>
> We thank you for appreciating our contributions. We sincerely hope our clarifications above have addressed your questions and can improve your opinion of our work.

---

> > ### Comment · Reviewer_61cM · 2025-04-07
> >
> > Thanks to the authors for their response.
> >
> > I do not have any further queries.

---

> > > ### Author Response · Authors · 2025-04-08
> > >
> > > Dear Reviewer 61cM,
> > >
> > > We are very happy to know that you have no further questions. Your invaluable review comments are immensely beneficial in improving the quality of our paper! Thank you!
> > >
> > > Best,
> > >
> > > The Authors

---

### Decision · Program_Chairs · 2025-05-01

**Decision:**

Accept (poster)

**Comment:**

This paper is recommended for unanimous acceptance by all reviewers, three of whom engaged with the authors’ responses. The work presents a novel and unified approach to both classificaiton and regression in test-time adaptation (TTA), with extensive evaluation performed on different tasks and architectures. The AC checked the paper and all reviewing material and concurred with acceptance.